# Identification of host proteins differentially associated with HIV-1 RNA splice variants

Rachel Knoener[1,2], Edward Evans III[2], Jordan T Becker[2], Mark Scalf[1], Bayleigh Benner[2], Nathan M Sherer[2]*, Lloyd M Smith[1]*

[1]Department of Chemistry, University of Wisconsin, Madison, United States; [2]McArdle Laboratory for Cancer Research and Institute for Molecular Virology, University of Wisconsin, Madison, United States

**Abstract** HIV-1 generates unspliced (US), partially spliced (PS), and completely spliced (CS) classes of RNAs, each playing distinct roles in viral replication. Elucidating their host protein 'interactomes' is crucial to understanding virus-host interplay. Here, we present HyPR-MS$_{SV}$ for isolation of US, PS, and CS transcripts from a single population of infected CD4+ T-cells and mass spectrometric identification of their in vivo protein interactomes. Analysis revealed 212 proteins differentially associated with the unique RNA classes, including preferential association of regulators of RNA stability with US and PS transcripts and, unexpectedly, mitochondria-linked proteins with US transcripts. Remarkably, >80 of these factors screened by siRNA knockdown impacted HIV-1 gene expression. Fluorescence microscopy confirmed several to co-localize with HIV-1 US RNA and exhibit changes in abundance and/or localization over the course of infection. This study validates HyPR-MS$_{SV}$ for discovery of viral splice variant protein interactomes and provides an unprecedented resource of factors and pathways likely important to HIV-1 replication.

*For correspondence:
nsherer@wisc.edu (NMS);
smith@chem.wisc.edu (LMS)

Competing interests: The authors declare that no competing interests exist.

## Introduction

HIV-1 uses the alternative splicing of a single primary RNA transcript to produce three major classes of viral RNA variants: unspliced (US), partially spliced (PS), and completely spliced (CS). The individual variants perform distinct roles during HIV-1 replication through dynamic interactions with specific viral and host proteins (*Coffin et al., 1997*). These protein 'interactomes' guide the RNA through required cellular pathways encompassing splicing, RNA nuclear export, mRNA translation, and packaging of full-length, US RNA genomes into progeny virions that assemble at the plasma membrane. Each HIV-1 splice variant performs a distinct function and is thus predicted to interface with a unique protein interactome.

HIV-1 gene expression is traditionally divided into two phases referred to as 'early' and 'late.' Early gene expression involves translation of auxiliary proteins Tat and Rev as well as the accessory protein Nef from CS transcripts. Tat and Rev localize to the nucleus where Tat facilitates viral transcription and Rev mediates nuclear export of intron-retaining US and PS transcripts. Late gene expression is marked by translation of the US transcript to synthesize Gag and Gag-Pol capsid proteins and translation of PS transcripts to generate Envelope glycoproteins as well as the Vpu, Vpr, and Vif immunomodulatory factors.

HIV-1 splicing generates vast numbers of splice variants, with over 50 proposed to be physiologically significant (*Emery et al., 2017*; *Ocwieja et al., 2012*; *Purcell and Martin, 1993*; *Vega et al., 2016*). The locations of splice donor and acceptor sites (*Sertznig et al., 2018*; *Vega et al., 2016*), the identities of several *cis*- and *trans*-regulatory elements (*Mahiet and Swanson, 2016*; *Sertznig et al., 2018*; *Stoltzfus, 2009*), and the transcript and protein product abundances needed

for efficient viral replication (*Cullen, 1991*; *Karn and Stoltzfus, 2012*; *Weinberger et al., 2005*) are still topics of intensive investigation toward the development of antiviral therapies.

Previous works have shown that the HIV-1 splice variant classes interact differentially with both viral and host proteins. For example, HIV-1 US and PS transcripts hijack the cellular XPO1 (also known as CRM1)-mediated nuclear export pathway through the activities of Rev and a *cis*-acting RNA structure known as the Rev-response element (RRE) (*Pollard and Malim, 1998*). Rev multimerizes on the RRE and recruits XPO1 to form a functional RNA export complex (*Bai et al., 2014*; *Daugherty et al., 2008*; *Daugherty et al., 2010*; *DiMattia et al., 2016*; *DiMattia et al., 2010*; *Fang et al., 2013*). This is in contrast to CS transcripts that do not require Rev and recruit components of the NXF1/NXT1 export machinery, similar to the bulk of cellular fully spliced mRNAs. A second example is HIV-1 genome packaging wherein US transcripts are packaged into virions due to favored interactions between the Gag polyprotein and a structured RNA packaging signal known as 'psi' in the 5'-untranslated region of the US transcript (*Berkowitz et al., 1993*; *Lever et al., 1989*; *Luban and Goff, 1994*). These binding sites are lost in PS and CS transcripts due to splicing (*Purcell and Martin, 1993*). Additional host factors have been implicated as RNA interactors regulating US, PS, and CS RNA expression and packaging (*Bolinger and Boris-Lawrie, 2009*; *Freed and Mouland, 2006*; *Jin and Musier-Forsyth, 2019*; *Mbonye and Karn, 2014*; *McLaren et al., 2008*; *Meng and Lever, 2013*; *Swanson and Malim, 2006*). However, the list is far from complete.

In the current study, we describe a novel strategy for efficient isolation of the three major HIV-1 splice variant classes from a single population of natively infected CD4+ T-cells and define distinct US, PS, and CS in vivo RNA-protein interactomes using mass spectrometry. We identify over 200 proteins differentially associated with the US, PS, and CS HIV splice variant pools, 102 of which are previously unkown HIV-1-host interactors. Gene-specific siRNA knockdown (KD) of 121 host proteins indicated more than 80 are effectors of HIV-1 RNA regulation. We further demonstrate, using fluorescence microscopy, several instances of identified host protein co-localization with HIV RNA and changes to the single-cell abundance and/or subcellular distribution of several identified host proteins over the course of HIV-1 infection. Collectively, we detail a powerful new approach for probing virus-host interactions and use it to expose new host factors with apparent roles in the HIV-1 replication cycle.

## Results

### Purification of HIV-1 splice variant classes

We recently described HyPR-MS (*Hy*bridization *P*urification of *R*NA-Protein Complexes Followed by *M*ass *S*pectrometry), a strategy to identify the in vivo protein interactomes of specific viral RNAs, lncRNAs, and mRNAs (*Knoener et al., 2017*; *Spiniello et al., 2018*; *Spiniello et al., 2019*). Here, we present HyPR-MS$_{sv}$, a strategy that expands the capabilities of HyPR-MS to differentiate in vivo protein interactomes for multiple splice variants (SV) derived from a single primary transcript and isolated from a single-cell population. Applied here, we purified the three major classes of HIV-1 splice variants (US, PS, and CS) from a single population of infected Jurkat CD4+ T-cells, then identified and characterized their protein interactomes using mass spectrometry.

To preserve in vivo viral RNA-protein complexes prior to cell lysis, Jurkat cultures were treated with formaldehyde at 48 hr post-infection (h.p.i.) (multiplicity of infection [MOI] of ~1 infectious unit per cell). The Jurkat cell line was chosen because it is a well-characterized CD4+ T-cell line previously confirmed to support replication of the HIV-1$_{NL4-3}$ reporter virus used for this study (*Knoener et al., 2017*). To isolate the US, PS, and CS RNA pools, three biotinylated capture oligonucleotides (COs) were designed complementary to three distinct regions of the HIV RNA genome: intron-1 (unique to US), intron-2 (present in both US and PS), and the 3'-exon (present in US, PS, and CS) (*Figure 1A*, *Supplementary file 1*). Cell lysates were first depleted of the US HIV RNA through hybridization to the intron-1 CO, followed by its capture with streptavidin-coated magnetic beads, and subsequent release using toehold-mediated oligonucleotide displacement. Additional hybridization, capture and release steps were subsequently repeated iteratively using first the intron-2 CO and then the 3'-exon CO for isolation of the PS and CS RNA pools, respectively. Once purified, proteins cross-linked to each isolated HIV RNA class were identified by mass spectrometry (overview in *Figure 1B*).

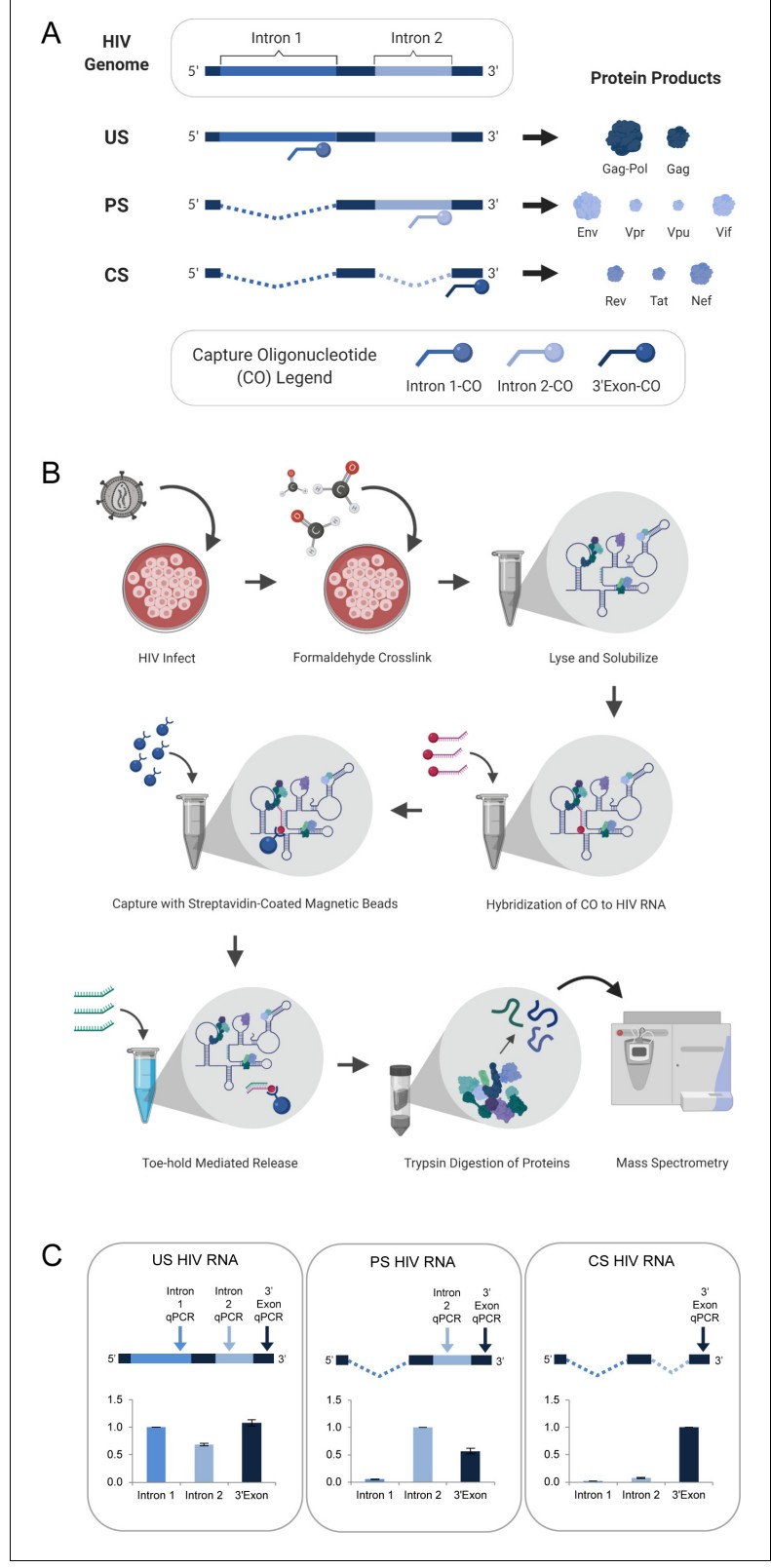

**Figure 1.** HyPR-MS for purification of HIV splice variant interactomes. (A) Capture oligonucleotides (COs) were designed to complement specific regions of the HIV genome to make possible the isolation of the three HIV splice variant classes from a single- cell lysate. (B) Overview of *Hy*bridization *P*urification of *R*NA-Protein Complexes Followed by *M*ass *S*pectrometry (HyPR-MS$_{SV}$) procedure. (C) Purification of the HIV splice variant

*Figure 1 continued on next page*

*Figure 1 continued*

classes was verified using RT-qPCR assays specific to regions in intron 1, intron 2, and 3'-exon. The intensity data is normalized to the intron 1 assay for unspliced (US) capture, the intron 2 assay for partially spliced (PS) capture, and the 3'-exon assay for completely spliced (CS) capture. Error bars are the standard deviation for three biological replicates. Figures A and B created with BioRender.com.

The online version of this article includes the following source data and figure supplement(s) for figure 1:

**Source data 1.** qPCR data for (A) capture specificity, (B) enrichment, and (C) capture efficiency calculations.

**Figure supplement 1.** RT-qPCR confirmation of enrichment for HIV splice variants and efficiency of capture.

---

RT-qPCR assays specific to intron 1, intron 2, and the 3'-exon (*Supplementary file 1*) were used to determine RNA capture specificity and efficiency. For three biological replicates of the US, PS, and CS captures, the magnitude of amplification using each qPCR assay confirmed strong capture specificity for the desired splice variant class relative to the other two classes (*Figure 1C*, *Figure 1— source data 1*). Enrichment of each HIV transcript over a cellular control transcript (*GAPDH*) was >100-fold (*Figure 1—figure supplement 1A*, *Figure 1—source data 1*). Capture efficiency (the amount of each transcript depleted from the lysate after capture) was >70% for each variant (*Figure 1—figure supplement 1B*, *Figure 1—source data 1*).

## Elucidation of unique protein interactomes for each HIV-1 splice variant class

To identify host proteins differentially interacting with the US, PS, and CS RNA pools, we isolated the in vivo cross-linked HIV RNA variants from three biological replicate experiments of $5 \times 10^7$ infected Jurkat cells; with each replicate generated from a separate set of cultured cells and virus preparation. Interacting proteins from the US, PS, and CS capture samples were purified, analyzed by bottom-up mass spectrometry, then identified and quantified using search and label-free quantitation algorithms (*Cox and Mann, 2008*; *Tyanova et al., 2016*). In all, 926 proteins were identified in at least two biological replicates of all three HIV splice variant captures (*Supplementary file 2*). More than two-thirds (633) of these common interactors were previously identified in at least one prior study identifying proteins associated with cellular polyadenylated mRNA (*Hentze et al., 2018*; *Queiroz et al., 2019*; *Supplementary file 2*), and thus likely represent RNA-associated proteins with general roles in mRNA processing, transport, and translation. These common interactors also featured several proteins previously implicated in HIV-1 replication including host factors regulating RNA transport and translational initation (e.g., NCBP1, DHX9, DDX3, EIF4G, and PABP) (*Boeras et al., 2016*; *Bolinger et al., 2010*; *Soto-Rifo et al., 2013*; *Stake et al., 2015*; *Yedavalli et al., 2004*), known HIV splicing factors (e.g., HMGA1, HNRNPA1, HNRNPAB, HNRNPH, HNRNPF, SRSF1, SRSF2, SRSF3, SRSF6, SRSF7, TRA2B, and U2AF2) (*Dlamini and Hull, 2017*; *Mahiet and Swanson, 2016*; *Sertznig et al., 2018*; *Stoltzfus and Madsen, 2006*), RNA nuclear export and transport proteins (e.g., ABCE1, RAB11A, RANBP2, and XPO1) (*Friedrich et al., 2011*), and proteins implicated in HIV-1 virus particle assembly (e.g., AP-2, PDCD6IP [ALIX], STAU2, UPF1, and VPS4) (*Friedrich et al., 2011*; *Meng and Lever, 2013*). Furthermore, several common interactors matched proteins previously identified in RNA capture screens that had used partial segments of HIV-1 RNA as bait, including 27 of 32 proteins identified by Kula et al., 32 of the 41 identified by Marchand et al., and 93 of the 121 identified by Stake et al. (*Supplementary file 2*; *Kula et al., 2011*; *Marchand et al., 2011*; *Stake et al., 2015*).

The primary goal of this study, however, was to use HyPR-MS$_{SV}$ to differentiate subsets of proteins preferentially associated with one or more of the individual splice variant classes. Therefore, we conducted three pairwise comparisons: US vs PS, US vs CS, and PS vs CS. Using the Student's t-test and a permutation-based false discovery rate (FDR) of 5%, we identified 212 proteins that differentially interacted with one or more of the HIV splice variant classes: 101, 93, and 68 proteins in the US, PS, and CS captures, respectively (*Figure 2—source data 1*). Hierarchical clustering was used to organize the 212 proteins into a heatmap for visualization (*Figure 2A*). The associated gene tree indicates the extent of similarity in 'interaction profiles' for the proteins shown; the length of the branches directly correlates with the degree of similarity. This analysis revealed clusters of proteins elevated for each individual class as well as proteins common to members of two HIV splice variant

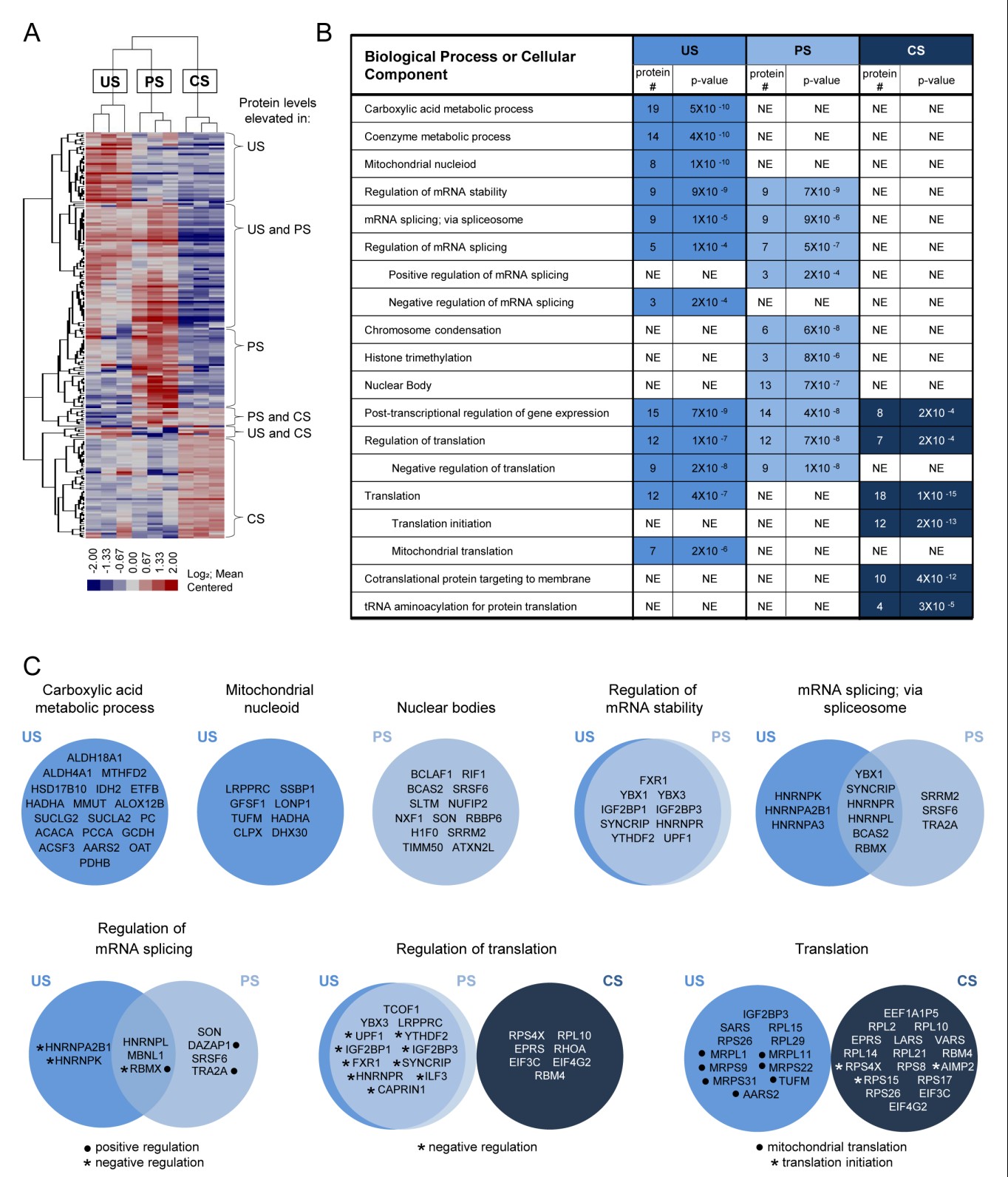

**Figure 2.** Determination and analysis of HIV splice variant protein interactomes. (**A**) Heatmap depicts relative intensities for each of the 212 proteins (rows) in each of the three biological replicates of the unspliced (US), partially spliced (PS), and completely spliced (CS) (columns) differential interactomes. (**B**) Condensed list of gene ontology (GO) biological process or cellular component terms enriched in each of the HIV splice variant interactomes. The 'protein #" column indicates the number of proteins in the interactome that are annotated with the biological process indicated. The

*Figure 2 continued on next page*

*Figure 2 continued*

'p-value' column indicates the likelihood that the proteins of the biological process are present in each interactome by random chance and were provided by GO term enrichment software (*Mi et al., 2017*). A lower p-value suggests non-random over-representation of a biological process. 'NE' = not enriched. (C) Venn diagrams of proteins annotated for biological processes or cellular components enriched in the splice variant differential interactomes.

The online version of this article includes the following source data for figure 2:

**Source data 1.** Pairwise comparisons of mass spectrometric data for US, PS, and CS RNA interactomes.

**Source data 2.** Gene ontology term enrichment analysis.

---

classes. The most abundant of these were proteins preferentially associated with both the US and PS HIV transcripts but not the CS pool (45 proteins) (*Figure 2A*).

We used these interactome data to infer biological pathways potentially relevant to the regulation of each splice variant class. Using gene ontology (GO) term enrichment algorithms (*Mi et al., 2017*), we evaluated each interactome for enrichment of proteins involved in specific biological processes. This analysis revealed over-representation of several GO terms in the interactome of each splice variant class; some common to more than one class (*Figure 2B*, *Figure 2—source data 2*). Notable among these were nine proteins, known to regulate RNA stability, associated with both the US and PS, but not CS, transcripts (FXR1, YBX1, YBX3, IGF2BP1, IGF2BP3, SYNCRIP, HNRNPR, YTHDF2, and UPF1). Proteins involved in mRNA splicing, and the regulation thereof, were also elevated in the US and PS relative to the CS capture samples (YBX1, SYNCRIP, HNRNPR, HNRNPL, BCAS2, RBMX, and MBNL1) but with less congruence. A subset of these proteins were elevated only in the US capture samples (HNRNPA2B1 and HNRNPK; negative regulators of splicing) or PS capture samples (DAZAP1 and TRA2A; positive regulators of splicing), but not in both. PS captures were also exclusively enriched for proteins found in nuclear bodies. For cytoplasmic activities, proteins involved in translation were highly enriched in both the US (12 proteins) and CS (18 proteins) interactomes. Interestingly, however, while the CS interactome included translation initiation proteins (as may be expected), the US interactome was enriched for proteins linked to mRNA translation in the mitochondria. Cellular component GO term enrichment analysis further revealed 45 mitochondrion proteins enriched in the US RNA interactome; eight are mitochondrial nucleoid proteins and several have known roles in the carboxylic acid metabolic process, a GO term also over-represented in the US RNA interactome (*Figure 2C*, *Figure 2—source data 2*).

## Validation of HyPR-MS$_{SV}$ defined HIV-1 RNA interactors using RNA silencing

To determine their potential relevance to HIV-1 gene expression, 121 host proteins identified by HyPR-MS$_{SV}$ were targeted for siRNA KD (*Supplementary file 3*) and passed cell viability criteria in HEK293T cells (*Figure 3—source data 1*). The HEK293T cell line was selected here due to its extensive use in studies of HIV-1 expression and its compatibility with siRNA transfection experiments (*König et al., 2008*). Following KD, cells were infected with a two-color HIV-1 virus engineered to report single-cell levels of viral US (Gag-cyan fluorescent protein [CFP]) and CS (mCherry) gene expression (*Figure 3A and B*; *Knoener et al., 2017*). Relative to a scrambled siRNA control, statistically significant changes (p-value<0.05) to early (CS) and/or late (US) gene expression were observed for a remarkable 69% (84 of 121) of the targeted host genes (*Figure 3C*; *Supplementary file 4*). The KD of 33 host proteins affected the expression of US and CS protein products in the same direction (either both increased or both decreased) and with approximately the same magnitude. By comparing mCherry:CFP fluorescence ratios for each protein KD to the negative control, we determined that CS and US protein expression were differentially affected by KD of 51 host proteins; for 26 of the proteins the expression changes were in the same direction but with different magnitudes; for 18 only the expression of the US RNA protein product was affected; and for seven only the expression of the CS RNA protein product was affected (*Figure 3C*, *Supplementary file 4*). Based on the direction of the changes in HIV-1 gene expression (increased or decreased), we categorized 71 host genes as putative 'positive' effectors and 16 as putative 'negative' effectors (*Figure 3C*, *Supplementary file 4*). Interestingly, of the 16 negative effectors, 10 were implicated in

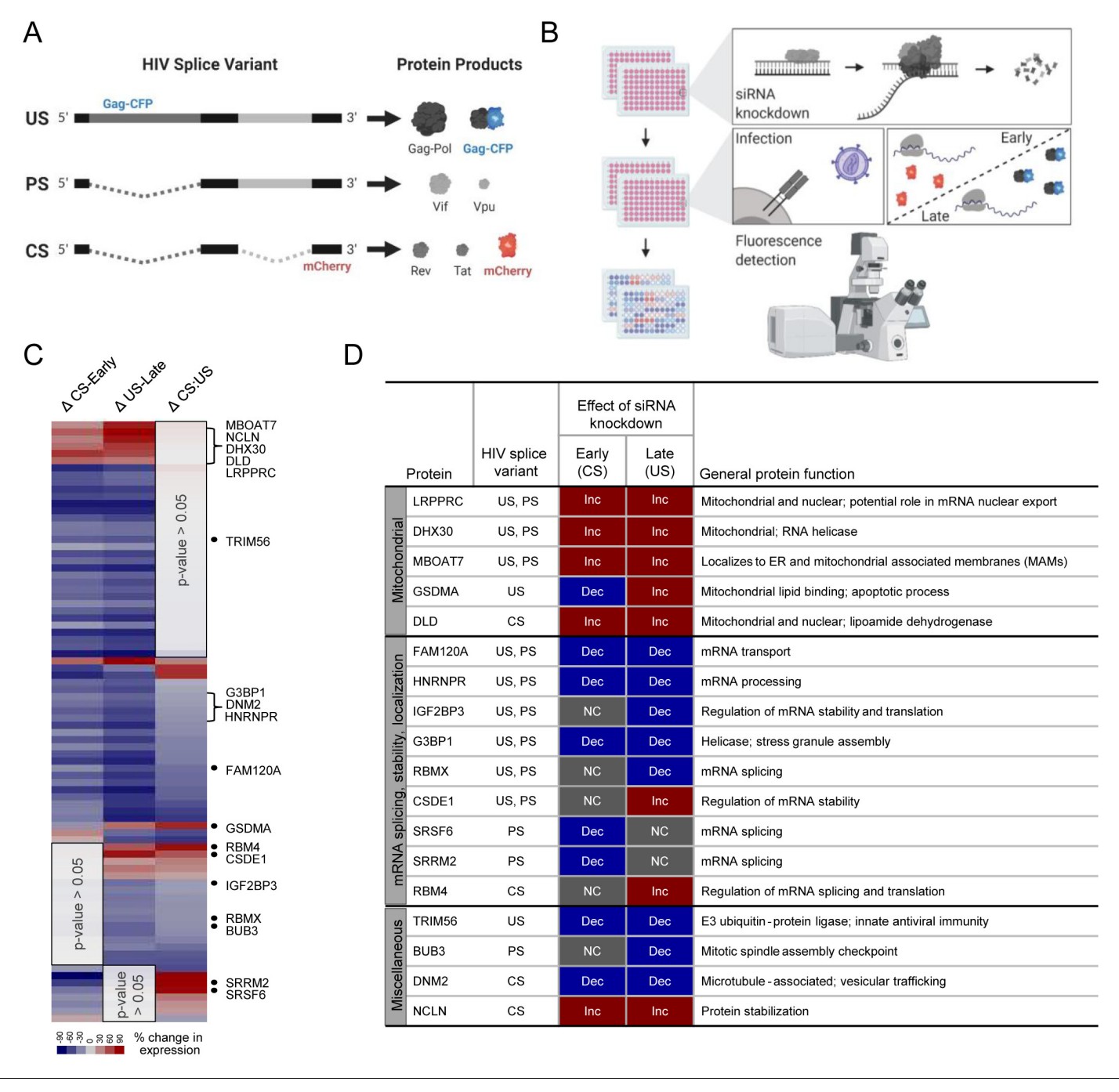

**Figure 3.** Screen for host protein effects on early and late HIV gene expression. (**A**) The HIV-1 reporter virus expresses mCherry from the *nef* locus as a completely spliced (CS) RNA 'early' stage reporter and Gag fused to cyan fluorescent protein (CFP) as a unspliced (US) RNA 'late' stage reporter. (**B**) In 96-well plates, 293 T-cells engineered to stably express an F-actin-YFP fusion protein as a host gene control (293T-ACT-YFP cells) were transfected with gene-specific siRNAs, incubated for 48 hr to initiate knockdown (KD), and then transfected with siRNAs for a second time prior to infection with the HIV reporter virus at a multiplicity of infection (MOI) of ~1. Cells were fixed at 48 hr post-incubation. Fluorescence microscopy was used to quantify CFP and mCherry levels. (**C**) Heatmap of HIV gene expression changes after siRNA KD of host proteins, with 84 of 121 proteins showing statistically significant changes in early and/or late HIV gene expression (p-values<0.05). (**D**) Table summarizes *Hy*bridization *P*urification of *R*NA-Protein Complexes Followed by *M*ass *S*pectrometry (HyPR-MS) and siRNA KD results for 18 genes of interest wherein KD was confirmed to be significant based on either quantitative immunoblot or immunofluorescence. Figures A and B created with BioRender.com.

The online version of this article includes the following source data and figure supplement(s) for figure 3:

**Source data 1.** siRNA KD and fluorescence expression data analysis.

*Figure 3 continued*

**Figure supplement 1.** Western blots confirm efficacy of siRNA knockdown strategy.

mitochondria-associated pathways based on GO analysis; of those ten, nine were identified by HyPR-MS$_{sv}$ to preferentially interact with the US HIV RNA (*Figure 2—source data 2*, *Supplementary file 4*).

## HyPR-MS$_{SV}$ candidates co-localize with US HIV RNA at distinct subcellular locations

We selected a subset of 20 HyPR-MS$_{SV}$ identified host proteins for further validation studies. This subset was, in part, chosen based on the commercial availability of antibodies that allowed for immunoblot- and/or immunofluorescence (IF)-based detection of the host proteins (*Supplementary file 5*), and included five proteins linked to mitochondria (LRPPRC, DHX30, MBOAT7, GSDMA, and DLD; all negative effectors of US RNA gene expression), ten genes encoding proteins with functions related to mRNA processing, localization, and stability (FAM120A, HNRNPR, IGF2BP3, G3BP1, RBMX, CSDE1, SRSF6, SRRM2, RBM4, and RPL15, the majority of which were positive effectors of either US or CS gene expression), and five additional proteins that had not previously been linked to RNA regulation (TRIM56, BUB3, DNM2, DYNC1H1, and NCLN) (*Figure 3D*, *Supplementary file 4*). Fifteen of the 20 proteins were detected by immunoblot and siRNA KD was confirmed (31–95% relative to negative control siRNA) (*Figure 3—figure supplement 1*). IGF2BP3, SRRM2, DNM2, RPL15, and DYNC1H1 KDs were not confirmed by immunoblot (*Supplementary file 6*).

US HIV-1 RNA-protein interactions may commence as early as production of the nascent HIV transcript in the nucleus or as late as virus particle formation at the plasma membrane. To determine potential sites of interaction, we used three-color combined fluorescence in situ hybridization (FISH)/IF single-cell imaging to show host factor subcellular localization relative to US RNA and viral Gag proteins (*Figure 4A* and *Figure 4—figure supplements 1–3*). HeLa cells were chosen based on their microscopy-conducive size and shape (large and flat) and their common use to study HIV-1 gene expression (*Jouvenet et al., 2008*; *Pocock et al., 2016*). Cells were infected with an HIV-1 reporter virus (HIV-1 E-R-CFP) allowing for identification of infected cells and confirmation of specificity of the US RNA FISH probes (Stellaris FISH probe set specific to intron-1; *Supplementary file 7*) and Gag antibody (anti-p24Gag; *Supplementary file 5*). Host proteins were detected using the primary antibodies employed for our immunoblot analysis (*Supplementary file 5*). Seventeen of the 18 host proteins (all but DLD) were detected by IF and showed >40% decreases in IF signal after host protein siRNA KD. This imaging-based analysis also allowed verification of the efficacy of siRNA KD for three of the host proteins (IGF2BP3, SRRM2, and DNM2) that we had been unable to detect using immunoblot (*Supplementary file 8*).

FISH/IF was performed on HeLa cells 48 hr post-infection to localize US RNA, Gag, and each host protein. Analysis by single-cell fluorescence microscopy showed consistent co-localization of US HIV-1 RNA with 11 of the proteins (*Figure 4A* and *Figure 4—figure supplements 1–3*), with four (HNRNPR, RBMX, RBM4, MBOAT7) predominantly localized to the nucleus or near the nuclear membrane and seven (FAM120A, IGF2BP3, MOV10, TRIM56, DNM2, LRPPRC, CSDE1) predominantly localized to the cytoplasm in uninfected cells (*Figure 4—figure supplement 4*). In infected cells, we observed five recurrent US RNA-host protein co-localization phenotypes: (1) at nuclear puncta, (2) at puncta proximal to the nuclear membrane, (3) at cytoplasmic puncta, (4) at large, cytoplasmic complexes reminiscent of stress granules, and (5) at the plasma membrane (*Figure 4A* and *Figure 4—figure supplements 1–3*). In the nucleus, we typically observed one or two bright US RNA puncta per cell, consistent with prior reports describing sites of active HIV-1 transcription (*Puray-Chavez et al., 2017*). Puncta proximal to the nuclear membrane and cytoplasmic puncta were smaller, more numerous, and of lower intensity. Cytoplasmic granules were large with moderate intensity accumulations of US HIV RNA surrounded by or spotted with host protein. Plasma membrane puncta were variable in size and intensity and often co-localized with Gag, thus likely represent virion assembly sites (*Figure 4A* and *Figure 4—figure supplements 1–3*).

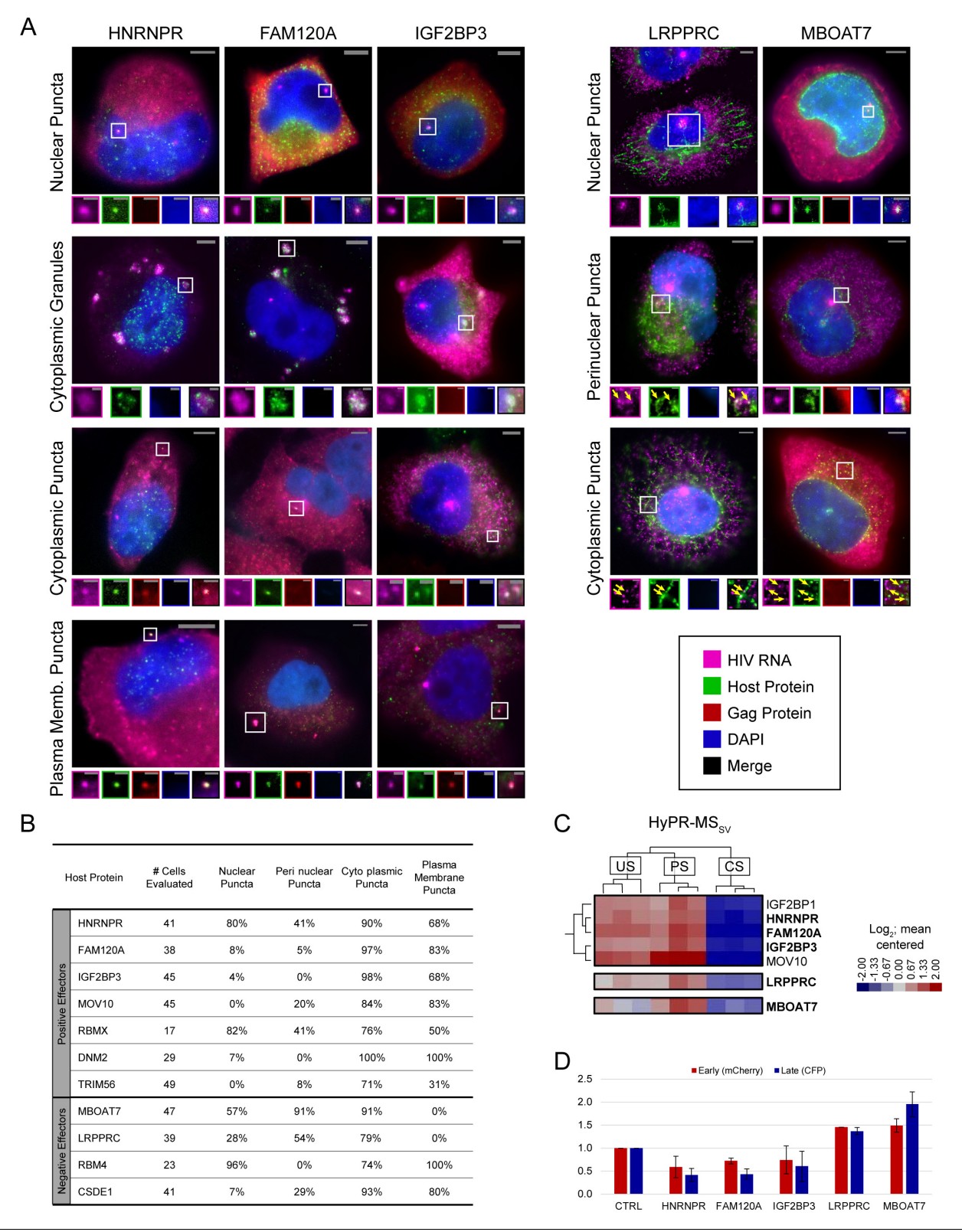

**Figure 4.** Unspliced (US) HIV RNA co-localizes with positive and negative effectors at multiple sites within the cell. (**A**) Representative images of co-localization phenotypes observed using fluorescence in situ hybridization/immunofluorescence (FISH/IF). For each, a merged image of a cell highlighting a site of co-localization (white square) is shown. Enlarged regions of interest (ROIs) of each fluorescence channel are displayed in the associated small panels to separate overlapping US HIV RNA, host protein, HIV Gag polyprotein, and DAPI signals. Some images were obtained from

*Figure 4 continued on next page*

*Figure 4 continued*

experimental replicates that did not include Gag IF and therefore do not include images from the corresponding channel. Note: Brightness and contrast settings were adjusted individually for each color channel of the images to effectively show co-localization. These settings may be different for the ROIs. (B) Table showing the frequency of observing a particular co-localization phenotype of US HIV RNA with each of 11 host proteins. Frequencies are displayed as the percentage of cells observed with each co-localization phenotype. (C) ROIs from the *Hy*bridization *P*urification of *R*NA-Protein Complexes Followed by *M*ass *S*pectrometry (HyPR-MS), hierarchically clustered heatmap (*Figure 2A*) showing the close relation of HNRNPR, FAM120A, and IGF2BP3 interaction profiles, each preferentially interacted with US and partially spliced (PS) HIV RNA. (D) Data for proteins of interest from the siRNA knockdown (KD) screen (*Figure 3—source data 1*). HIV gene expression decreases for HNRNPR, FAM120A, and IGF2BP3 upon KD with a greater decrease in late gene expression than in early. For mitochondria-related proteins LRPPRC and MBOAT7, HIV gene expression increases upon KD of the host protein.

The online version of this article includes the following source data and figure supplement(s) for figure 4:

**Source data 1.** Quantitation of host protein and HIV RNA co-localization phenotypes.
**Figure supplement 1.** Representative images of host proteins co-localizing with unspliced (US) HIV RNA.
**Figure supplement 2.** Representative images of host proteins co-localizing with unspliced (US) HIV RNA.
**Figure supplement 3.** Representative images of host proteins co-localizing with unspliced (US) HIV RNA.
**Figure supplement 4.** Representative images of host protein general cellular localization.

We quantified the frequency of each co-localization phenotype for 17–52 cells per antibody (*Figure 4B*, *Figure 4—source data 1*), excluding cytoplasmic granules that were only rarely observed. The data revealed that proteins that predominantly localize to the nucleus or proximal to the nuclear membrane (HNRNPR, RBMX, RBM4, MBOAT7) had a higher frequency of co-localization with HIV RNA at nuclear puncta (57–96%) relative to proteins that were predominantly localized to the cytoplasm (FAM120A, IGF2BP3, MOV10, DNM2, TRIM56, LRPPRC, CSDE1; 0–8%). Two proteins (HNRNPR and RBMX) co-localized frequently with HIV-1 US RNA at all four quantified sites (41–90%). All 11 host proteins co-localized with US RNA at small cytoplasmic puncta in a high percentage of cells (71–100%) with most (all but LRPPRC and MBOAT7) co-localizing with US RNA at the plasma membrane (31–100% of cells), generally with Gag also present (*Figure 4B*, *Figure 4—source data 1*).

## A subset of HyPR-MS$_{SV}$ candidates appear to associate with US RNAs from sites of transcription to the cytoplasm

Several HyPR-MS candidates (HNRNPR, FAM120A, IGF2BP3, RBMX, RBM4, CSDE1, DNM2, LRPPRC, MBOAT7) were observed to accumulate at bright US RNA nuclear puncta, suggesting that they associate with US RNA at or near sites of de novo transcription (*Figure 4—figure supplements 1–3*). Of these, HNRNPR, FAM120A, and IGF2BP3 were of particular interest because all three exhibited four US HIV RNA co-localization phenotypes (nuclear puncta, cytoplasmic granules, cytoplasmic puncta, and plasma membrane puncta) (*Figure 4B*); preferentially interacted with US and PS, but not CS, HIV RNA as determined by HyPR-MS$_{SV}$ (*Figure 4C*, *Supplementary file 4*), and showed similar trends in gene expression upon siRNA KD (both US and CS expression decreased, with a marginally greater decrease in US expression) (*Figure 4D*, *Supplementary file 4*). By contrast, LRPPRC, a protein shown to localize to the mitochondria as well as the nucleus (*Mili and Piñol-Roma, 2003*; *Ruzzenente et al., 2012*), and MBOAT7, a protein shown to localize to mitochondria-associated membranes (*Hirata et al., 2013*), localized to US HIV RNA nuclear puncta, at perinuclear puncta, and at cytoplasmic puncta but were not observed to co-localize with US RNA and Gag at the plasma membrane. Similar to HNRNPR/FAM120A/IGF2BP3, both LRPPRC and MBOAT7 were preferentially associated with US and PS, relative to CS, transcripts based on HyPR-MS$_{SV}$ analysis. However, unlike HNRNPR/FAM120A/IGF2BP3, each of these proteins were negative effectors of both US and CS HIV-1 gene expression (*Figure 4D*, *Supplementary file 4*). Interestingly, LRPPRC was detected not only near transcription sites but also in a trail-like pattern that extended to the periphery of the nucleus (*Figure 4A* and *Figure 4—figure supplements 1–3*) and MBOAT7 was observed at transcription sites, at smaller subnuclear US HIV RNA puncta, and with high frequency and abundance at US RNA puncta at or near the nuclear membrane (*Figure 4A* and *Figure 4—figure supplements 1–3*).

## HIV-1 infection alters the abundance and localization of several HyPR-MS$_{SV}$ identified proteins

The FISH/IF single-cell analyses of US HIV RNA, Gag, and host proteins also allowed for tracking of host factor responses to infection (*Figure 5*). For example, HNRNPR, generally a nuclear protein, was primarily localized to the nucleus of cells expressing no, or low amounts of, Gag and US RNA, but exhibited marked shifts from the nucleus to the cytoplasm in cells with high levels of Gag and US RNA expression (*Figure 5A*). Changes to MBOAT7 were also striking, with much higher levels of expression in cells with abundant Gag and US RNA than in uninfected or early infected cells (*Figure 5B*).

To further track HyPR-MS$_{SV}$ host factor changes, we plotted single-cell measurements of total Gag and total US HIV RNA and used the resulting inflection point to discriminate cells in 'early' and 'late' stages of HIV gene expression (*Figure 5C and D*; *Figure 5—source data 1*). We measured relative host protein abundances for 12 of these factors at these stages (HNRNPR, FAM120A, IGRF2BP3, LRPPRC, MBOAT7, CSDE1, DNM2, MOV10, RBM4, RBMX, SRRM2, TRIM56; *Figure 5E and F*, *Figure 5—source data 1*). In general, each host protein exhibited non-random, bimodal expression changes from 'early' and 'late' HIV gene expression (*Figure 5—figure supplements 1–4*, *Figure 5—source data 1*). For example, in early/uninfected cells, we observed linear increases in HNRNPR and MBOAT7 expression, positively correlating with the subtle increases in US HIV RNA expression (*Figure 5E and F*; slope m = 1.7, 0.9, respectively). However, in late cells, HNRNPR expression rose then fell again as per-cell US RNA increased, fitting a polynomial rather than linear trendline ($R^2$ = 0.690) (*Figure 5E*), while MBOAT7 expression levels plateaued (*Figure 5F*, slope m = −0.007).

A similar analysis was performed after image-based segmentation of cells into nuclear and cytoplasmic compartments to better discriminate the subcellular location in which host protein changes occurred (*Figure 5—figure supplements 1–4*, *Figure 5—source data 1*). For HNRNPR, the same trends were observed in the nucleus and cytoplasm as were seen for the total cell (*Figure 5—figure supplements 1–4*). For MBOAT7, nuclear expression plateaued as it did for total cell expression, but the cytoplasmic expression increased slightly as US RNA and Gag abundance increased (*Figure 5—figure supplements 1–4*). In all, the expression of each of the 12 host proteins showed significant correlation with the expression of US HIV RNA in at least one of the following sub-groups: early-nuclear, late-nuclear, early-cytoplasmic, late-cytoplasmic (*Figure 5—figure supplement 5*, *Figure 5—source data 1*). For HNRNPR and MBOAT7, the most evident differences were in cytoplasmic expression (cyto HNRNPR, median increase = 21%, p=0.055; cyto MBOAT7, median increase = 41%, p=3×10$^{-5}$) (*Figure 5G–J*, *Figure 5—figure supplements 1–4*, *Figure 5—source data 1*). In all, changes to nuclear or cytoplasmic abundance were observed for five host proteins (p-values<0.05; MBOAT7, TRIM56, RBMX, MOV10, and IGF2BP3) (*Figure 5K*, *Figure 5—figure supplements 1–4*, *Figure 5—source data 1*). Three of these proteins showed differences to total cellular expression (MBOAT7, RBMX, and TRIM56), with RBMX and TRIM56 only increasing in the cytoplasm. Two proteins did not show net differences in overall expression but exhibited statistically significant differences (p-value<0.05) in expression in the nucleus (MOV10) or the cytoplasm (IGF2BP3).

To identify potential host protein translocation events, we evaluated single-cell nuclear-to-cytoplasmic (nuc/cyto) ratios relative to US RNA abundance and looked for statistically significant differences in early and late cells (*Figure 5—figure supplements 1–4*, *Figure 5—source data 1*). HNRNPR nuc/cyto ratios ranged from 1 to 3 in early/uninfected cells but only ranged from 0.6 to 0.9 in late infected cells, exhibiting a negative correlation with US HIV RNA expression (*Figure 5L*). For MBOAT7, the nuc/cyto ratio ranged from 1.7 to 3.9 in early cells and 1.5 to 3.9 in late cells, with no significant correlation with US RNA expression for either phase (*Figure 5M*). However, overall nuc/cyto ratios were significantly lower for late cells relative to early cells for both proteins (median decrease = -52%; p=5×10$^{-11}$ and median decrease=-27%; p=7×10$^{-6}$, respectively) (*Figure 5N and O*). In all, the nuc/cyto ratios of six HyPR-MS candidate proteins showed notable changes to nuc/cyto ratio (HNRNPR, MBOAT7, TRIM56, SRRM2, RBMX, and RBM4); all with the exception of SRRM2, exhibiting relative increases to cytoplasmic abundance (*Figure 5P*, *Figure 5—source data 1*).

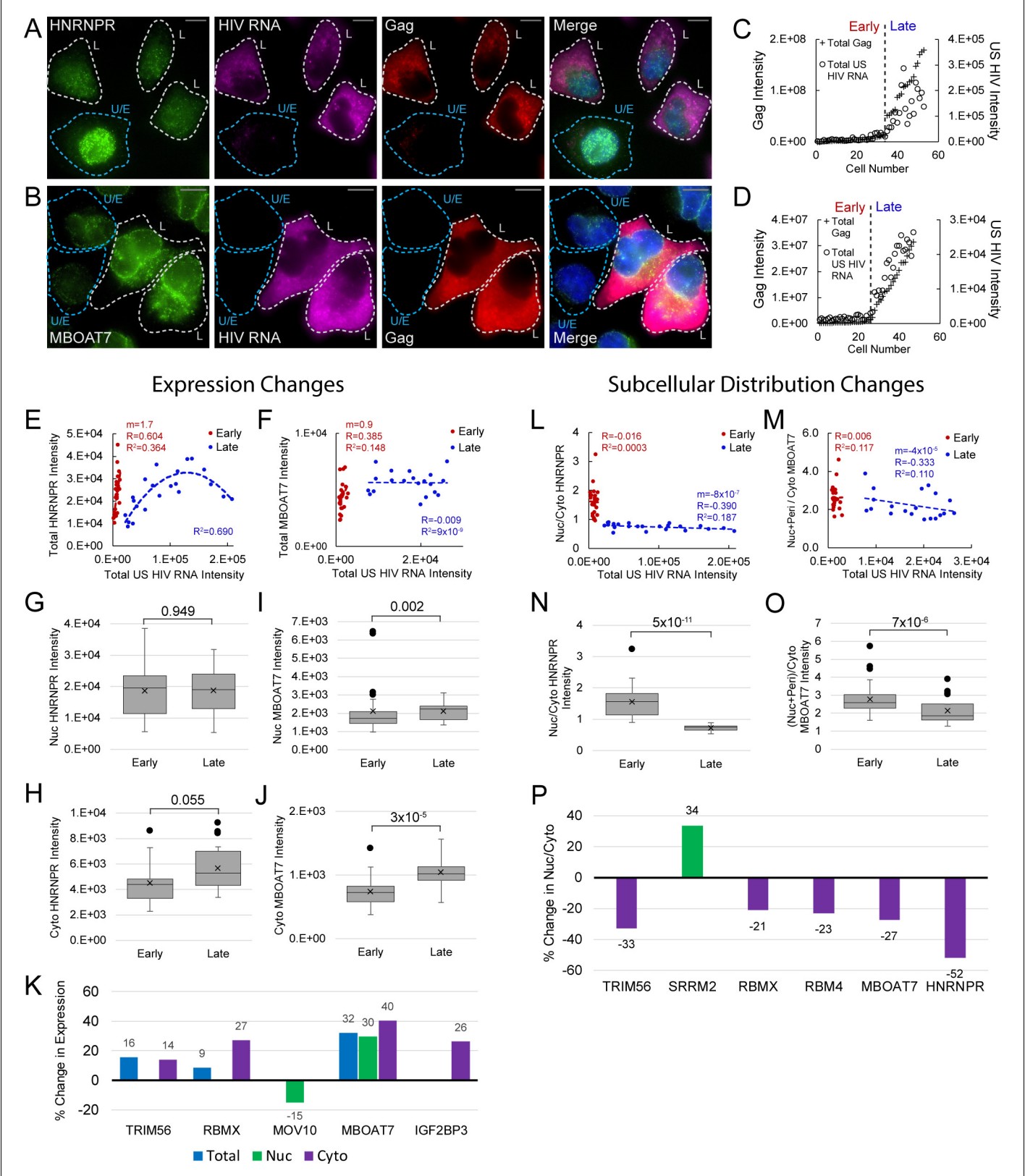

**Figure 5.** Host protein expression and cellular distribution. (**A**) HNRNPR cellular distribution appears to be different in uninfected/early stage infected cells (U/E; blue outlines) than in late stage infected cells (L; white outlines). (**B**) MBOAT7 expression appears greater in late stage infected cells than in uninfected/early stage infected cells. (**C–D**) Plots of cellular Gag-immunofluorescence (IF) and unspliced (US) HIV RNA-fluorescence in situ

*Figure 5 continued on next page*

*Figure 5 continued*

hybridization (FISH) intensities for cells analyzed for HNRNPR (**C**) and MBOAT7 (**D**). Cells prior to the inflection points in each plot are termed 'early cells' as they are either uninfected or at stages of HIV replication prior to late gene expression (i.e., high amounts of Gag in the cytoplasm). Cells after the inflection point are termed 'late cells' as they express high, IF-detectable levels of Gag in the cytoplasm. (**E–F**) For each cell in the early and late cell sub-groups, the intensity of US HIV RNA vs the intensity of HNRNPR (**E**) and MBOAT7 (**F**) is plotted. Linear or polynomial regressions ($R^2$) are fit to each early and late sub-group and the Pearson's correlation coefficient (**R**) calculated for linear regressions. This demonstrates the extent of correlation between US HIV RNA expression and the expression of each host protein. (**G–J**) A Student's t-test is applied to determine if the host protein intensities in early cells are significantly different from those in late cells in the nucleus and the cytoplasm. (**K**) The percent change in the median expression for all host proteins with early vs late p-values<0.05. Calculations were made for total cell, nuclear, and cytoplasmic differences. (**L–M**) Total cellular US HIV RNA intensities vs host protein nuclear to cytoplasmic (nuc/cyto) or nuclear plus perinuclear (nuc+peri)/cyto ratios. This demonstrates the extent of correlation of host protein cellular distribution with US HIV RNA expression. (**N–O**) A Student's t-test measures significant differences in the cellular distribution between the early and late cells for HNRNPR and MBOAT7. (**P**) The percent change in the median nuc/cyto or nuc+peri/cyto ratio for all host proteins with p-values<0.05. Purple indicates the late cells have a higher proportion of the host protein in the cytoplasm than do the early cells. Green indicates the late cells have a higher proportion in the nucleus.

The online version of this article includes the following source data and figure supplement(s) for figure 5:

**Source data 1.** Quantitation of expression and distribution changes of host proteins in early and late HIV infection.
**Figure supplement 1.** Host protein expression and cellular distribution.
**Figure supplement 2.** Host protein expression and cellular distribution.
**Figure supplement 3.** Host protein expression and cellular distribution.
**Figure supplement 4.** Host protein expression and cellular distribution.
**Figure supplement 5.** Summary of host protein expression and cellular distribution.

Taken together, these analyses demonstrated that many of the host factors identified by HyPR-MS_SV not only modulate HIV-1 gene expression (*Figure 3*) but also co-localize with HIV-1 US RNA (*Figure 4*) and respond during infection by increasing in abundance and/or undergoing alterations to subcellular distribution (*Figure 5*).

## Discussion

The variation in gene products encoded by the HIV-1 genome is largely achieved through regulated synthesis of a diverse RNA transcriptome. Deciphering the distinct cellular processes each splice variant undergoes and the host proteins involved is critical to understanding HIV-1 replication. Here, using HIV-1 as a relevant model system, we describe HyPR-MS_SV as a new tool that can be applied to elucidate distinct protein interactomes for distinct HIV-1 splice variant classes.

Isolation of the multiple HIV splice variant classes and comparative analysis of their differential protein interactors yielded a rich interactome resource valuable for studies of HIV-1 RNA metabolism. Notably, the protein interactomes of the three splice variant classes differ markedly from one another, presumably reflecting functional differences (*Figure 2*). We uncovered over 50 proteins that differentially impacted early and late HIV gene expression based on siRNA KD (*Figure 3*), mapped the cellular locations where several of the host proteins co-localized with US HIV RNA (*Figure 4*), and established a correlation between infection and altered levels of expression or subcellular localization for several host proteins (*Figure 5*). Combined, these results provide a roadmap for RNA-protein interactions potentially central to HIV replication (*Figure 6*).

In developing HyPR-MS_SV, we aimed to (1) ensure the relevance of protein interactors by only pursuing interactions that occur in cells (i.e., in vivo), (2) ensure versatility of the technique for broad applications wherein it is useful to differentiate between one or more RNA splice variants for comparative RNA-capture proteomics, and (3) determine, for the first time, the protein interactomes for the three major HIV splice variant classes. The first two goals were achieved by configuring HyPR-MS_SV to sequentially deplete specific classes of HIV RNA from the same pool of natively infected cell lysates using three independently targeted sets of short (~30 nt) biotinylated capture oligos (*Figure 1*). To our knowledge, all previous studies for discovery of HIV RNA protein interactors, with the exception of our prior study (*Knoener et al., 2017*), utilized synthetic viral RNAs as bait added to cellular lysates (*Marchand et al., 2011*; *Singh et al., 2016*; *Stake et al., 2015*) or viral constructs engineered to encode artificial RNA sequences for the purpose of RNA 'tagging' (e.g., MS2 loops) (*Kula et al., 2011*). While effective at identifying protein interactors, both of these strategies may complicate the interpretation of results by eliminating the cellular context of interactions (e.g.,

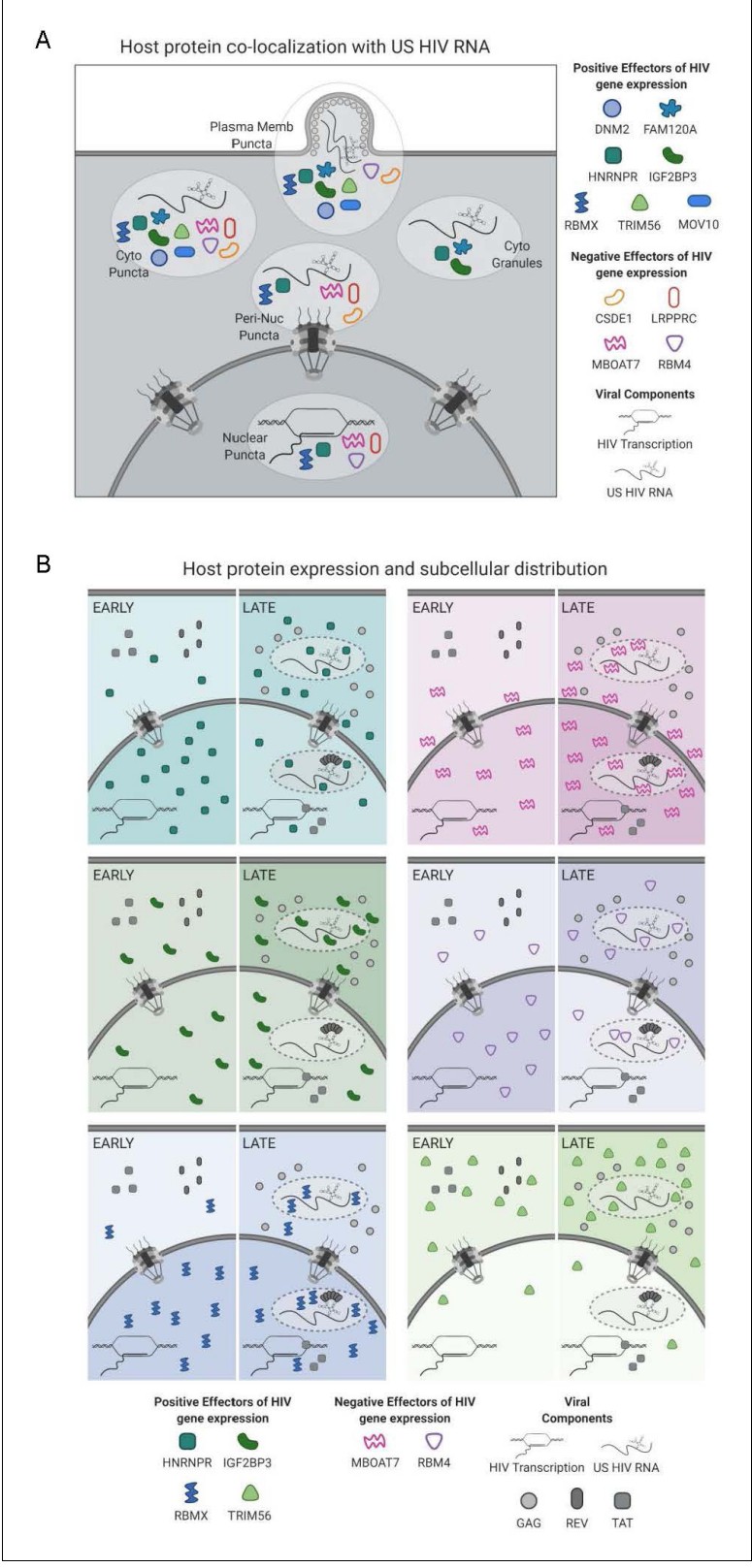

**Figure 6.** Models for subcellular co-localization, expression, and distribution of HIV-1 RNA-associated host proteins during infection. (A) Summary of cellular co-localization phenotypes observed by fluorescence in situ hybridization/immunofluorescence (FISH/IF) analysis of HIV-1 unspliced (US) RNA and select host proteins. Host proteins represented include both positive and negative effectors of HIV gene expression as was determined by

*Figure 6 continued on next page*

*Figure 6 continued*

siRNA knockdown. (**B**) Models of host protein changes in expression and cellular distribution from early to late HIV gene expression. Host protein quantities represented here show the general, but not exact, scale of changes in host protein abundance in the nucleus and cytoplasm over the course of HIV replication. The extent of shading in the nucleus or cytoplasm correlates with that same change in abundance. Figure created with BioRender.com. The online version of this article includes the following figure supplement(s) for figure 6:

**Figure supplement 1.** Heatmap summary of closely clustered proteins of interest.

interactions may occur at specific stages of the viral replication cycle) or by introducing non-native components that might interfere with native interactions. A majority of the proteins identified in the abovementioned studies were also identified by HyPR-MS$_{SV}$ (*Supplementary file 2*), providing validation of our strategy. Many of these proteins were differential interactors of the HIV RNA splice variant classes showing HyPR-MS$_{SV}$ provides the novel capability to resolve not only the stage at which a given host factor interacts with the viral RNA (i.e., time of cross-linking post-infection) but also the specific class of HIV-1 splice variant with which it associates (*Figure 2—source data 1*). Furthermore, this strategy can, in theory, be expanded to differentiate the protein interactomes of each individual protein-coding HIV-1 RNA, can be used to extract native RNA transcripts produced from any strain or infected cell type, and can easily be adapted to study other viruses or cellular RNA splice variants.

By isolating the HIV-1 splice variant classes, we were able to compare RNAs containing both shared and distinct sequences, and likely corresponding secondary, tertiary, and quaternary structures, to decipher how their protein interactors may consequently differ. We validated the sequential capture of US, PS, and CS HIV RNAs using RT-qPCR and showed at least 200-fold specificity relative to cellular RNAs and at least 10-fold specificity for the splice variant class of interest in each capture. Over 900 proteins were detected in the interactomes of all three HIV splice variant classes (*Supplementary file 2*); many of these are likely 'general' RNA regulatory factors as they were found associated with cellular polyadenylated RNAs in prior studies (*Hentze et al., 2018*; *Queiroz et al., 2019*). However, many of these common interactors included RNA regulatory proteins already demonstrated to play important roles in HIV-1 replication (e.g., DHX9, DDX3, SR proteins, and ABCE1) (*Boeras et al., 2016*; *Bolinger et al., 2010*; *Friedrich et al., 2011*; *Mahiet and Swanson, 2016*; *Soto-Rifo et al., 2013*; *Stoltzfus and Madsen, 2006*; *Yedavalli et al., 2004*) or were identified in prior pull-down studies (*Kula et al., 2011*; *Marchand et al., 2011*; *Stake et al., 2015*).

Based on statistical analysis, however, only ~200 proteins were associated preferentially with only a subset of HIV-1 splice variant classes at 48 h.p.i. Among these were proteins specific to the US, PS, or CS HIV RNAs as well as a large number of proteins enriched in both the US and PS HIV RNA captures (*Figure 2A* and *Supplementary file 4*). Overall, of the 212 proteins identified as HIV-1 RNA interactors, 25 had been previously shown to associate with US HIV RNA (*Knoener et al., 2017*) and 57 with viral proteins (Gag, Gag-Pol, Tat, Rev) that are key regulators of HIV-1 RNA regulation (*Oughtred et al., 2019*; *Supplementary file 2*). Based on siRNA KD, at least 48 represent potential new host regulatory factors (*Supplementary file 4*). Using GO term enrichment analysis we showed that several biological processes and cellular components are over-represented in each splice variant sub-group (*Figure 2B*, *Figure 2—source data 2*), suggesting cellular pathways that may be uniquely involved in the processing of a subset of HIV splice variants. Notable was enrichment of proteins related to the regulation of mRNA stability and splicing in the US and PS interactomes and proteins related to mitochondrial gene expression and organization in the US interactome.

We speculate that a subset of the factors common to US and PS transcripts are likely to play roles in the Rev/XPO1-driven nuclear export pathway or in promoting cytoplasmic utilization of intron-retaining mRNAs. By contrast, a subset of the factors preferentially associated with US RNA, but not PS or CS transcripts, may play roles in US RNA genome packaging and/or the virus assembly pathway. To focus on US and PS RNA nuclear export, we examined a cluster of 16 proteins with closely related HIV splice variant interaction profiles which preferentially associated with US and PS, but not CS, transcripts (*Figure 6—figure supplement 1*, *Supplementary file 4*). Eleven of these host proteins were known stress granule components and/or had functions in splicing. Ten were determined to be positive effectors of late gene expression, six of which affected late gene expression significantly more than early. Within this group, a subcluster of three proteins (HNRNPR, FAM120A, and

IGF2BP3) exhibited markedly similar HyPR-MS RNA interaction profiles (*Figure 4C*), siRNA KD effects (*Figure 4D*), and subcellular localization patterns (*Figure 4A*). HNRNPR and IGF2BP3, as well as IGF2BP1 and YBX1 which also cluster with this group (*Figure 6—figure supplement 1*), were previously identified as components of IGF2BP1-ribonucleoprotein granules (IMP1-granules); cytoplasmic granules that contain and confer stability to mRNAs that have not yet been translated (*Jønson et al., 2007*). HNRNPR was also shown to stabilize and facilitate subcellular localization of RNA (*Briese et al., 2018*; *Reches et al., 2016*). Intriguingly, YBX1, IGF2BP1, and HNRNPR have previously been suggested to have roles in HIV replication: YBX1 was shown to stabilize HIV US RNA and enhance virus production (*Jung et al., 2018*; *Mu et al., 2013*), overexpression of IGF2BP1 was shown to reduce HIV infectivity through its interaction with Gag (*Zhou et al., 2008*), and HNRNPR was shown to interact with HIV Rev (*Hadian et al., 2009*). By contrast, FAM120A has not previously been linked to viruses but has been shown to protect RNAs from Ago2-mediated degradation through the RNA-induced silencing complex (RISC) (*Kelly et al., 2019*), which frequently serves in an antiviral role (*Eckenfelder et al., 2017*; *Harvey et al., 2011*). Our functional analysis showed a decrease in late HIV gene expression upon KD of all five of these proteins (IGF2BP1, YBX1, HNRNPR, IGF2BP3, and FAM120A; *Supplementary file 4*), consistent with shared roles for these clustered host proteins as positive regulators of US HIV RNA transport and/or stability. That HNRNPR, IGF2BP1, and FAM120A were co-localized with Gag at plasma membrane punctae (*Figure 4A*), likely assembling virions, may suggest that that these proteins play a role in the virion assembly pathway.

Analysis of the HyPR-MS and siRNA KD screen data also revealed a trend for mitochondria-linked proteins that interacted with US and/or PS HIV RNA and served as negative effectors of HIV gene expression. Of particular interest were LRPPRC and MBOAT7 because they both preferentially interacted with US and PS HIV RNA, were categorized as negative effectors of late gene expression, and were detected co-localizing with US RNA both in the nucleus and the cytoplasm. LRPPRC was previously shown to localize to the nucleus and to mitochondria as a putative effector of RNA metabolism in both locations (*Mili and Piñol-Roma, 2003*; *Ruzzenente et al., 2012*). One study showed that nuclear LRPPRC directly interacted with XPO1, eIF4E, and a signature RNA secondary structure found in a subset of cellular RNAs (*Volpon et al., 2017*), features similar to how Rev and the RRE are known to drive US and PS RNA export. Interestingly, another study implicated LRPPRC in HIV-1 replication but as affecting the pre-integration stages (*Schweitzer et al., 2012*). By contrast, MBOAT7 is an intramembrane protein and acyltransferase that incorporates polyunsaturated fatty acids into phosphatidylinositol (*Lee et al., 2008*; *Lee et al., 2012*) and has not previously been implicated in viral or cellular RNA metabolism. However, in addition to perinuclear and ER localization, MBOAT7 has been reported to localize to mitochondrial associated ER membranes which bridge the ER to the mitochondria to regulate antiviral signaling through the mitochondrial antiviral-signaling viral RNA sensor (*Hirata et al., 2013*). Based on our combined results, we hypothesize that both LRPPRC and MBOAT7 link US RNA transport to mitochondrial signaling pathways capable of dampening HIV-1 late gene expression.

Notably, many metabolic and enzymatic proteins were identified as US HIV RNA interactors (*Figure 2B and C*). Prior studies have indicated that many metabolic enzymes, including ALDH18A1, HSD17B10, IDH2, and SUCLG1 identified here, 'moonlight' as RNA binding proteins. (*Beckmann et al., 2015*; *Castello et al., 2015*; *Cieśla, 2006*; *Hentze et al., 2018*). The precise role in RNA regulation for most of these enzymes is still unknown; however, it is thought that they may regulate RNA expression directly through binding, or conversely, the RNA could influence the enzyme's metabolic function by outcompeting the enzyme's typical substrate (*Castello et al., 2015*; *Hentze et al., 2018*). In our siRNA screen, the KD of HSD17B10 and IDH2 both resulted in a statistically significant change in HIV late gene expression but not early gene expression, with HSD17B10 acting as a negative effector and IDH2 as a positive effector (*Supplementary file 3*). This suggests that metabolic proteins may play an important role in HIV gene expression, but further investigation is need to decipher the unique role of each.

It should be noted that the HyPR-MS interactome determination, siRNA KD screen, and RNA-protein co-localization and expression analysis were completed using three different cell lines: Jurkat, 293T, and HeLa, respectively. Each cell line was selected for its previously demonstrated utility in HIV-1 research and its compatibility with each technology being used here. We acknowledge that differences in gene expression profiles or protein functions among these cell lines result in some

limitations in the conclusions potentially derived from these data. Here, we have limited our analysis to proteins that were shown to be HIV-1 RNA interactors or effectors in the analyses done using all three cell lines. To overcome some of these limitations, future work will include conducting similar screens in primary CD4+ T-cells and monocytes/macrophages using more relevant R5-tropic HIV-1 founder strains for elucidation of innate immune signaling host factors involved in HIV-1 RNA biology.

Overall, this study describes and validates a powerful new biochemical approach for deep interrogation of the complex interplay of viral and cellular RNA and protein factors during viral infection (*Figure 6*). Using siRNA KD and single-cell imaging experiments, we also generated a catalog of host protein candidates for positive and negative regulation of HIV-1 gene expression. Interestingly, a subset of proteins (HNRNPR, RBM4, and RBMX) were frequently observed both at US RNA transcription sites as well as at putative sites of virus particle assembly, suggesting that these factors may be capable of strong, persistent association with viral RNA throughout the entire productive phase (*Figure 6A*). The changes seen in subcellular distribution of HNRNPR are consistent with a role in stability and nuclear export of intron-retaining HIV-1 transcripts while the expression changes and localization of IGF2BP3 support a role in cytoplasmic US RNA transport and stability (*Figure 6B*). Finally, we identified a set of host factors linked to mitochondria (including LRPPRC and MBOAT7) that may represent new effectors of HIV-1 antiviral surveillance.

## Materials and methods

### Cell lines

Jurkat E6 cells are T lymphocytes established from the peripheral blood of a 14-year-old male with acute T-cell leukemia and were obtained through the NIH AIDS Reagent Program, Division of AIDS, NIAID, NIH: Jurkat Clone E6-1 from Dr. Arthur Weiss (cat# 177) (*Weiss et al., 1984*). The cells were cultured in RPMI media supplemented with 10% fetal bovine serum and 1% L-glutamine-penicillin-streptomycin in roller bottles rotated at three rotations per minute (rpm) at 37°C in 5% $CO_2$. A cell density of $1 \times 10^6$ cells/mL of media was maintained by regular quantification.

HEK293T cells are human embryonic kidney cells and were obtained from ATCC (CRL-11268). The cells were cultured in Dulbecco's modified Eagle's medium (DMEM) supplemented with 10% fetal bovine serum and 1% L-glutamine-penicillin-streptomycin at 37°C in 5% $CO_2$. This cell line was authenticated by morphology and are G418 resistant.

Human 293 T-cells obtained from ATCC were modified to stably express YFP-tagged F-actin (ACT-YFP) and were cultured in DMEM supplemented with 10% fetal bovine serum, 1% L-glutamine, and 1% penicillin-streptomycin.

HeLa cells obtained from ATCC were cultured in DMEM media supplemented with 10% fetal bovine serum and 1% L-glutamine-penicillin-streptomycin at 37°C in 5% $CO_2$.

All cell lines were validated as mycoplasma-negative using a PCR assay developed by *Uphoff and Drexler, 2002*; *Uphoff and Drexler, 2004*.

### HIV-1 virion production

$2.5 \times 10^6$ HEK293T cells were plated in 10 cm tissue-culture-treated dishes in 10 mL media, then transfected using polyethylenimine with 1 µg of DNA plasmid expressing the G envelope glycoprotein from vesicular stomatitis virus (VSV-G) and 9 µg of plasmid DNA encoding the full-length NL4-3 molecular clone of HIV-1 bearing inactivating mutations in *env*, *vpr*, and either (1) expressing a CFP reporter from the *nef* reading frame (HIV-1 E-R-CFP) (*Adachi et al., 1986*; *Becker and Sherer, 2017*) or (2) expressing mCherry in the nef ORF and three copies of CFP, in tandem, between the matrix and capsid ORFs of Gag (E-R- Gag-CFP mCherry/nef) (*Knoener et al., 2017*). At 24 hr post-transfection, media was replaced with 4 mL fresh media. At 48 hr post-transfection, culture supernatants were harvested, filtered through a sterile 0.45 µm syringe filter, and frozen at −80°C. Dose of HIV-1 E-R-CFP viral inoculum required for effective infection was determined in small-scale infection titrations in Jurkat cells.

## HyPR-MS analysis

### Jurkat cell infections

$1 \times 10^8$ Jurkat cells in 25 mL RPMI, 25 mL viral inoculum (HIV-1 E-R-CFP) in DMEM, and polybrene (concentration 10 µg/mL) were combined and incubated in a rotating roller bottle. After 3 hr, culture volume was increased to 300 mL using RPMI media and incubated at 3 rpm for 45 hr. Infection was confirmed to be >90% by visualizing CFP expression via epifluorescence microscopy. Cells were centrifuged at 1500 rpm for 10 min, washed three times with phosphate buffered saline (PBS), then cross-linked by resuspending in 0.25% formaldehyde and incubated at room temperature for 10 min. Cross-linked cells were washed once with PBS, then resuspended in 100 mM Tris-HCl, and incubated at room temperature for 10 min to quench formaldehyde. Cells were washed twice more in 1xPBS, pelleted by centrifugation, and frozen at −80°C.

### Cell lysis

Jurkat cell pellets were resuspended on ice in lysis buffer (469 mM LiCl, 62.5 mM Tris HCl, pH 7.5, 1.25% LiDS, 1.25% Triton X-100, 12.5 mM Ribonucleoside Vanadyl Complex, 12.5 mM DTT, 125 U/mL RNasin Plus, 1.25× Halt Protease Inhibitors) to a final cell concentration of $5 \times 10^6$ cells/mL. Cells were lysed by frequent vortexing for 10 min, keeping the cells on ice between vortexes.

### HIV-1 RNA splice variant hybridization and capture

Each capture replicate used $5 \times 10^7$ cells. Three biological replicates of each splice variant capture were conducted for this analysis. The HIV-1 US, PS, and CS RNAs were each purified from the Jurkat cell lysate by three sequential and separate hybridization and capture events: US followed by PS followed by CS HIV RNA. The amounts of biotinylated COs and streptavidin-coated magnetic beads for each hybridization and capture are listed in *Supplementary file 1*. The appropriate concentrations of biotinylated COs were added to the Jurkat cell lysates, and the final concentration of lysis buffer (375 mM LiCl, 50 mM Tris, 1% LiDS, 1% Triton X-100, 10 mM RVC, 10 mM DTT, 100 U/mL RNasin Plus, 1× Halt Protease Inhibitors) was obtained by addition of nuclease free water. The samples were then incubated at 37°C for 3 hr with gentle nutation. Streptavidin-coated magnetic Speedbeads were washed three times with wash buffer (375 mM LiCl, 50 mM Tris, 0.2% LiDS, 0.2% Triton X-100) prior to addition and nutation for 1 hr at 37°C with the hybridization samples. Using a magnet, the beads were collected to the side of each tube, and the lysate was removed and transferred to a clean tube for the next hybridization and capture. The beads were then washed two times, for 15 min each, at 37°C with a volume of wash buffer five times the volume of the original aliquot of beads used for capture (i.e., 5× bead volume) then one time for 5 min at room temperature with a 5× bead volume of release buffer (100 mM LiCl, 50 mM Tris, 0.1% LiDS, 0.1% Triton X-100).

### Release of HIV RNA from beads

The beads for the US, PS, and CS RNA captures were individually resuspended in a 3× bead volume of release buffer. The appropriate amount of release oligonucleotide (*Supplementary file 1*) was added and the bead mixture was nutated at room temperature for 30 min. Using a magnet to collect the beads to the side of the tube, the supernatant containing the released RNA-protein complexes was transferred to a clean tube. The resulting sample was divided into two aliquots: 2% for RT-qPCR analysis and 98% for mass spectrometric analysis.

### RNA extraction and reverse transcription

Two percent by volume of each release sample was incubated overnight at 37°C with 1 mg/mL proteinase K, 4 mM CaCl₂, and 0.2% LiDS to remove the proteins. The RNA was then extracted from the samples using TriReagent per manufacturer's protocol and precipitated in 75% ethanol, with 2 µL of GlycoBlue, at −20°C for at least 2 hr. The RNA was pelleted by centrifugation at 20,800 g and 4°C for 15 min, the pellet was washed with 75% ethanol, centrifuged at 20,800 g and 20°C for 15 min, then resuspended in 15 µL of nuclease free water; 10 µL of the purified RNA was used for reverse transcription (High Capacity cDNA Reverse Transcription Kit, Applied Biosystems) per the manufacturer's protocol. The procedures described here were also performed on HIV-1 E-R-CFP virus inoculum for isolation and analysis of a semi-purified standard of the US HIV RNA. The isolated

RNA was quantified by NanoDrop analysis, serially diluted, reverse transcribed, and then used for a standard calibration curve for qPCR analysis.

## qPCR analysis

The 20 µL reverse transcription product was diluted with 20 µL of nuclease free water and analyzed using sequence-specific qPCR primers and probes (*Supplementary file 1*) and Roche LightCycler 480 Probes Master Mix for relative quantitation of the US, PS, and CS HIV transcripts and human GAPDH mRNA. Purified HIV-1 E-R-CFP plasmid was quantified by NanoDrop analysis, serially diluted, and then used as a standard calibration curve for qPCR analysis.

## Protein purification and trypsin digestion

Ninety-eight percent by volume of each capture sample was processed using an adapted version of eFASP (*Erde et al., 2014*) for purification of proteins. Amicon 50 kDa MWCO filters and collection tubes were passivated by incubating overnight in 1% CHAPS and then rinsed thoroughly with mass spectrometry grade water. Each release sample was brought to a final concentration of 8 M urea and 0.1% deoxycholic acid (DCA) then passed through the filter in 500 µL increments by centrifugation for 10 min at 14,000 g. RNA-protein complexes were trapped in the filter and the eluent passed through to a collection tube for discarding. In the same manner (addition of solution followed by centrifugation), the following passages were conducted: (1) three passages of 400 µL of exchange buffer (8 M urea, 0.1% DCA, 50 mM Tris pH 7.5), (2) incubation for 30 min with 200 µL of reducing buffer (8 M urea, 20 mM DTT), (3) incubation for 30 min, in the dark, with alkylation buffer (8 M urea, 50 mM iodoacetamide, 50 mM ammonium bicarbonate), and (4) three passages of 400 µL of digestion buffer (1 M urea, 50 mM ammonium bicarbonate, 0.1% DCA). Finally, the sample remaining in the filter was brought to 100 µL with digestion buffer, the filter was transferred to a clean, passivated collection tube, and 1 µg of trypsin added to the filter for digestion. The filter-collection tube containing the sample was sealed with parafilm to prevent evaporation during incubation overnight at 37°C. Following digestion, the filter collection tube was centrifuged for 10 min at 14,000 g; 50 µL of 50 mM ammonium bicarbonate was added to the filter followed by centrifugation at 14,000 g for 10 min. This step was repeated once to ensure the collection of the entire peptide sample. The 200 µL peptide sample was then brought to 1% trifluoroacetic acid (TFA) followed by addition of 200 µL of ethyl acetate. The sample was vortexed for 1 min, then centrifuged at 15,800 g for 2 min. The top layer was aspirated and discarded and extraction with 200 µL ethyl acetate was repeated two times. The aqueous layer was then dried using a Savant SVC-100H SpeedVac Concentrator and the sample resuspended in 150 µL 0.1% TFA. For removal of salts from the sample, a C18 solid-phase extraction pipette tip was first conditioned with 70% acetonitrile (ACN), 0.1% TFA, and then equilibrated with 0.1% TFA. The peptide sample was then loaded onto the C18 solid phase by repeated passing of the 150 µL sample over the cartridge. The C18 extraction pipette tip was then rinsed with 0.1% TFA 10 times followed by peptide elution in 150 µL 70% ACN, 0.1% TFA. The samples were then dried using the SpeedVac Concentrator and reconstituted in 95:5 $H_2O$:ACN, 0.1% formic acid.

## Mass spectrometry of peptides

The samples were analyzed using an HPLC-ESI-MS/MS system consisting of a high-performance liquid chromatography (nanoAcquity, Waters) set in line with an electrospray ionization (ESI) Orbitrap mass spectrometer (LTQ Velos, ThermoFisher Scientific). A 100 µm id × 365 µm od fused silica capillary micro-column packed with 20 cm of 1.7 µm diameter, 130 Å pore size, C18 beads (Waters BEH), and an emitter tip pulled to approximately 1 µm using a laser puller (Sutter Instruments) was used for HPLC separation of peptides. Peptides were loaded on-column with 2% acetonitrile in 0.1% formic acid at a flow-rate of 400 nL/min for 30 min. Peptides were then eluted at a flow-rate of 300 nL/min over 120 min with a gradient from 2% to 30% acetonitrile, in 0.1% formic acid. Full-mass profile scans were performed in the FT orbitrap between 375 and 1500 m/z at a resolution of 120,000, followed by MS/MS HCD scans of the 10 highest intensity parent ions at 30% relative collision energy and 15,000 resolution, with a mass range starting at 100 m/z. Dynamic exclusion was enabled with a repeat count of one over a duration of 30 s. The Orbitrap raw files were analyzed using MaxQuant (version 1.5.3.30) (*Cox and Mann, 2008*) and searched with Andromeda (*Cox et al., 2011*) using the combined Uniprot (*Breuza et al., 2016*) canonical protein databases for human and HIV-1 and

supplemented with common contaminants (downloaded June 8, 2016). Samples were searched allowing for a fragment ion mass tolerance of 20 ppm and cysteine carbamidomethylation (static) and methionine oxidation (variable). A 1% FDR for both peptides and proteins was applied. Up to two missed cleavages per peptide were allowed and at least two peptides were required for protein identification and quantitation. Protein quantitation was achieved using the sum of the peptide peak intensities for each protein of each biological replicate and capture sample type. The peak intensities of HIV capture samples were normalized by the total peak intensity of all HIV capture samples and the same was done for scrambled capture samples.

### MS data analysis

To determine the differential interactomes of the HIV-1 splice variants, pairwise comparisons (US vs PS, US vs CS, PS vs CS) were statistically analyzed with the Student's t-test and a permutation-based FDR (5% threshold) using Perseus software (*Tyanova et al., 2016*; *Figure 2—source data 1*). Proteins that met this threshold in at least one pairwise comparison were then hierarchically clustered using Cluster software (*de Hoon et al., 2004*). For each protein, the intensities across all capture samples were mean centered and an uncentered correlation with centroid linkage clustering algorithm was applied. The clustering software provided a mathematical representation of protein-profile similarity, and TreeView (*Saldanha, 2004*) was used to graphically visualize the similarities and differences in the protein profiles of the differential interactomes. GO analysis, using PANTHER (*Mi et al., 2017*), of the proteins statistically elevated in each individual splice variant capture was evaluated for enrichment of terms in the categories of biological processes and cellular component (*Figure 2— source data 2*).

## siRNA KD screen

### Cell culture, KD, and infection

The virus used for determining the effect of gene-specific siRNA KD on early and late HIV-1 gene expression was a two-color fluorescent HIV-1 reporter virus (E-R-Gag-CFP mCherry/nef). This virus expresses mCherry in the nef ORF and three copies of CFP, in tandem, between the matrix and capsid ORFs of Gag, in a similar but expanded manner as previously done (*Hendrix et al., 2015*; *Holmes et al., 2015*; *Mergener et al., 1992*). This virus allows for the screening of early (mCherry; CS gene products) and late (CFP; US gene products) gene expression. Stocks of viral inoculum were produced in 293 T-cells by transfecting plasmids encoding E-R-Gag-CFP mCherry/nef with plasmid psPAX2 (an HIV-1 packaging helper plasmid that improves titer) and VSV-G. Human 293 T-cells stably expressing ACT-YFP were cultured in DMEM supplemented with 10% fetal bovine serum, 1% L-glutamine, and 1% penicillin-streptomycin. All cell incubations during the siRNA KD process were done at 37°C and 5% $CO_2$ in a humidified incubator. Approximately $5 \times 10^3$ cells were plated in 24 wells of a 96-well culture plate and incubated 24 hr; then the media was replaced with 125 µL of anti-biotic-free DMEM; 0.875 µL of DarmaFECT transfection reagent in 25 µL of Opti-MEM was mixed with 25 µL of Opti-MEM containing 4.4 pmol of gene-specific siRNA (*Supplementary file 3*), incubated at room temperature for 20 min, then added to the appropriate well of the 96-well plate. The final, in-well concentration of each siRNA was 25 nM. After 4 hr of incubation, the media was replaced with fresh DMEM media and the cells were incubated overnight. Twenty-four hours post-transfection, the cells were lifted from the bottom of the well by gentle pipetting, divided equally into two wells, incubated for another 24 hr, then again each well containing cells was divided equally into two wells (now a total of four wells for each gene-specific siRNA KD). The cells were allowed to adhere for 2–4 hr, then a second siRNA transfection as described above was conducted in all four wells. Four hours post-transfection, the siRNA containing media was replaced with fresh media. Additionally, in two of the four wells, polybrene was added to the media (final concentration of 2µg/ mL) followed by the HIV-1 reporter virus (E-R- Gag-3xCFP mCherry/nef) inoculum in DMEM. After 24 hr the media was exchanged for fresh media and 48 h.p.i. the cells were washed with PBS and fixed for 12 min using 4% paraformaldehyde in PBS, then stored at 4°C in PBS until imaged. Two biological replicates, each consisting of two technical replicates of infected and two technical replicates of uninfected cells, were obtained for each siRNA targeted gene. Biological replicates are defined as full siRNA KD procedures, from cell plating to cell fixation, performed on different days.

## Imaging

Imaging experiments were performed on a Nikon Ti-Eclipse inverted wide-field epifluorescence deconvolution microscope (Nikon Corporation). Images were collected using an Orca-Flash 4.0 C11440 (Hamamatsu Photonics) camera and Nikon NIS Elements software (version 4.20.03) using Nikon 4x/0.13 (Plan Apo) objective lense and the following excitation/emission filter set ranges (wavelengths in nanometers): 418-442/458-482 (CFP), 490-510/520-550 (YFP), and 555-589/602-662 (mCherry).

## Image processing and HIV gene expression quantitation

Images were processed and analyzed using FIJI/ImageJ2 (*Rueden et al., 2017*). For each well, only cell monolayers were used for quantitation of fluorescence. Cell viability for each gene-specific siRNA KD was assessed using the ACT-YFP marker expression in the 293 T-cell line. Wells that had YFP fluorescence detected within +/- 1.5 standard deviations of the plate mean were considered acceptable for further analysis. CFP and mCherry fluorescence for each well was normalized to the YFP fluorescence for the gene-specific siRNA and negative control siRNA (included in each 96-well plate). The Student's t-test calculation was performed to determine if a statistically significant change in CFP or mCherry expression was detected between each gene-specific siRNA KD and the negative control siRNA KD (*Figure 3—source data 1*; p-value<0.05).

## Western blot validation

HEK293T cells, with and without gene-specific siRNA KD, were lysed in 1x radioimmunoprecipitation assay buffer (10 mM Tris-HCl [pH 7.5], 150 mM NaCl, 1 mM EDTA, 0.1% sodium dodecyl sulfate (SDS), 1% Triton X-100, 1% sodium deoxycholate) and sonicated. Samples were then boiled for 10 min in 2x dissociation buffer (62.5 mM Tris-HCl [pH 6.8], 10% glycerol, 2% SDS, 10% β-mercaptoethanol), run on SDS-PAGE 10% polyacrylamide gels, and transfered to nitrocellulose membranes (0.2 µM pore size). Immunoblotting was performed as previously described (*Becker and Sherer, 2017*; *Behrens et al., 2017*; *Garcia-Miranda et al., 2016*) using the primary and secondary antibodies detailed in *Supplementary file 5*.

## Co-localization and expression quantitation

### Cell culture and infection

HeLa cells cultured in DMEM in eight-well Ibidi plates were infected with HIV-1 attenuated virus (E-R-CFP), described above, by adding polybrene to each well at a final concentration of 2 µg/mL followed by the virus inoculum. Media was exchanged with fresh media 24 h.p.i. The cells were fixed 48 h.p.i. by washing with PBS, incubating with 3.7% formaldehyde for 10 min at room temperature, then washing three times with PBS. Cells were then made permeable by incubating with 0.2% Triton X-100 for 15 min at room temperature and washing three times with PBS. Endogenous RNases were then deactivated by incubating with 0.1% diethyl pyrocarbonate (DEPC) in PBS for 15 min, removing the solution then incubating again with fresh 0.1% DEPC in PBS for 15 min; the cells were then washed three times with PBS and stored at 4°C.

### IF labeling

All IF steps were conducted at room temperature. Blocking buffer (Western blocking buffer, Roche) was added to each well, incubated for 30–60 min, then removed. Cells were then incubated with fresh blocking buffer containing the appropriate primary antibodies at designated concentrations (*Supplementary file 5*) for 60 min followed by four, 5 min, washes with blocking buffer. Blocking buffer containing appropriate concentrations of the secondary antibodies and DAPI stain (*Supplementary file 5*) were then incubated with the cells for 40 min followed by four, 5 min, washes with PBS. Finally, the cells were fixed with 3.7% formaldehyde for 10 min followed by three washes with PBS.

### Fluorescence in situ hybridization

The FISH protocol was conducted using Stellaris designed hybridization probes (*Supplementary file 7*) and Stellaris FISH reagents. All FISH steps were conducted in the dark. Cells were washed with FISH Wash Buffer A for 5 min at room temperature. FISH Hybridization Buffer containing 12.5 nM

FISH probes was added to each well and incubated at 37°C for 4 hr. The cells were then washed twice with FISH Wash Buffer A for 30 min at 37°C, then washed once with FISH Wash Buffer B for 5 min at room temperature.

### Order of protocols

The performance of each primary antibody was dependent on the order that the FISH and IF protocols were performed. For some protein/antibody pairs (DNM2, HNRNPR, FAM120A, MBOAT7, MOV10, RBM4, RBMX) the IF signal was superior if the IF was conducted prior to FISH. For other antibodies (CSDE1, LRPPRC, TRIM56) the IF signal was superior if the IF was conducted after FISH. For G3BP1 and IGF2BP3, either order was fine. If a Gag primary antibody was used, it was added at the same time as the host protein primary antibody. The protocols for each procedure remained consistent, the order in which they were done was only reversed.

### HIV RNA, Gag, and host protein single-cell imaging

Single-cell imaging experiments were performed on a Nikon Ti-Eclipse inverted wide-field epifluorescence deconvolution microscope (Nikon Corporation). Images were collected using an Orca-Flash 4.0 C11440 (Hamamatsu Photonics) camera and Nikon NIS Elements software (version 4.20.03) using Nikon 60x (N.A. 1.40; Plan Apo) or 100× (N.A. 1.45; Plan Apo) objective lenses and the following excitation/emission filter set ranges (wavelengths in nanometers): 405/470 (DAPI), 430/470 (CFP), 490/525 (AlexaFluor488), 585/610 (CAL Fluor Red 590), 645/705 (AlexaFluor647). Images were generally acquired in z-stacks containing various numbers of images along the z-axis of the cells. Images were processed and analyzed using FIJI/ImageJ2 (*Rueden et al., 2017*). All z-frames within a z-stack were examined for instances of co-localization; however, the fluorescence from only a single z-frame was used to produce co-localization images. For determining HIV RNA, Gag, and host protein expression differences in cells, four z-frames were merged additively for fluorescence quantitation of each component.

## Quantitation of HIV RNA, Gag, and host protein IF

Fluorescence for each channel (HIV-RNA, Gag protein, and each host protein) was quantified using FIJI/ImageJ2 (*Rueden et al., 2017*). Nuclear and total cellular fluorescence were measured by drawing perimeters around the nucleus (defined by DAPI staining) and the entire cell (defined by Gag protein fluorescence in late stage cells or autofluorescence in uninfected/early stage cells), then using FIJI quantitation tools to measure the fluorescence within each drawn perimeter. Cytoplasmic fluorescence was calculated by subtracting nuclear fluorescence from total cell fluorescence and the nuc/cyto ratio was calculated by dividing the nuclear fluorescence by the cytoplasmic fluorescence (*Figure 5—source data 1*). For determining correlation of host protein expression with HIV gRNA expression, cells with outlier values in total HIV gRNA fluorescence were excluded from the dataset. An outlier here is defined as a value that is more than 1.5 interquartile ranges (IQRs) below the 1st quartile (Q1) or above the 3rd quartile (Q3). IQR is defined as (Q3–Q1), with Q3 and Q1 calculated using the quartile function in Excel (*Figure 5—source data 1*). The Pearson's R value was calculated, excluding outliers, to determine correlation of US HIV RNA and host protein fluorescence expression in the nucleus, cytoplasm, total cell, and for the nuc/cyto ratios using the CORREL function in Excel. $R^2$ values were calculated using the chart tools in Excel (*Figure 5—source data 1*).

For determining host protein expression and distribution changes, outliers were determined, as described above, for host protein expression values in early and late cells in four categories: nuclear, cytoplasmic, total cellular, and nuc/cyto ratio. To determine if 'early' and 'late' cells showed statistically significant differences in host protein expression, a Student's t-test, excluding outliers, was used to determine a p-value. The percent change in each category was calculated using the mean values for each category within early and late cells.

## Quantification and statistical analysis

Statistical methods are described in the appropriate 'Materials and Methods' section or figure captions for all data analyses.

## Acknowledgements

This study was supported by National Institutes of Health [R01AI110221, U54AI150470 to NMS, R01CA193481 to LMS, T32CA009135 to ELE]; the Greater Milwaukee Foundation's Shaw Scientist Program [to NMS]; a UW-Madison UW2020 Infrastructure Award [to NMS]; two National Science Foundation Graduate Research Fellowships [DGE-1256259 to JTB and BEB], an OVCGRE Dissertation Completion Fellowship [to JTB]; an Advance Opportunity Fellowship from the UW-Madison SciMed/GRS program [to ELE]. Any opinions, findings, and conclusions or recommendations expressed in this material are those of the authors and do not necessarily reflect the views of the National Science Foundation.

## Additional information

### Funding

| Funder | Grant reference number | Author |
| --- | --- | --- |
| National Institutes of Health | R01AI110221 | Nathan M Sherer |
| National Institutes of Health | U54AI150470 | Nathan M Sherer |
| National Institutes of Health | R01CA193481 | Lloyd M Smith |
| National Institutes of Health | T32CA009135 | Edward Evans III |
| National Science Foundation | DGE-1256259 | Jordan T Becker<br>Bayleigh Benner |
| University of Wisconsin-Madison | UW2020 Infrastructure Award | Nathan M Sherer |
| Greater Milwaukee Foundation | Shaw Scientist Program | Nathan M Sherer |
| Office of the Vice Chancellor for Research and Graduate Education, University of Wisconsin-Madison | Dissertation Completion Fellowship | Jordan T Becker |
| University of Wisconsin-Madison | Advance Opportunity Fellowship (SciMed/GRS program) | Edward Evans III |

The funders had no role in study design, data collection and interpretation, or the decision to submit the work for publication.

### Author contributions

Rachel Knoener, Conceptualization, Data curation, Formal analysis, Validation, Investigation, Visualization, Methodology, Writing - original draft, Project administration, Writing - review and editing; Edward Evans III, Jordan T Becker, Mark Scalf, Bayleigh Benner, Investigation, Writing - review and editing; Nathan M Sherer, Lloyd M Smith, Conceptualization, Resources, Supervision, Funding acquisition, Methodology, Project administration, Writing - review and editing

### Author ORCIDs

Rachel Knoener (ID) https://orcid.org/0000-0002-2787-1098
Jordan T Becker (ID) http://orcid.org/0000-0002-0239-5443
Bayleigh Benner (ID) http://orcid.org/0000-0002-6266-5740
Nathan M Sherer (ID) https://orcid.org/0000-0001-9974-236X
Lloyd M Smith (ID) https://orcid.org/0000-0002-6652-8639

### Decision letter and Author response

Decision letter https://doi.org/10.7554/eLife.62470.sa1
Author response https://doi.org/10.7554/eLife.62470.sa2

# Additional files

## Supplementary files

- Supplementary file 1. Human immunodeficiency virus type 1 (HIV-1) splice variant capture oligonucleotide and qPCR assay sequences and genomic locations. Related to *Figure 1* and *Figure 1—figure supplement 1*.

- Supplementary file 2. Proteins identified by *Hy*bridization *P*urification of *R*NA-Protein Complexes Followed by *Mass Spectrometry* (HyPR-MS) to be 'common protein interactors,' or proteins identified in at least two of three biological replicates of each splice variant class capture, and proteins identified to be 'differential protein interactors.'. Includes comparison of common protein interactors to a meta-analysis of studies identifying general mRNA interactors. Also includes comparison of proteins previously determined to interact with human immunodeficiency virus type 1 (HIV-1) RNAs.

- Supplementary file 3. siRNA sequences used for gene-specific knockdown screen. Related to *Figure 3*.

- Supplementary file 4. Summary of human immunodeficiency virus type 1 (HIV-1) splice variant interactome MS data and siRNA knockdown (KD) expression changes data. Related to *Figures 2* and *3*.

- Supplementary file 5. Antibodies used for immunoblots and immunofluorescence. Related to *Figures 4* and *5* and *Figure 4—figure supplements 1–3*.

- Supplementary file 6. Western blot quantitation, with and without siRNA knockdown (KD), with and without human immunodeficiency virus type 1 (HIV-1) infection. Related to *Figure 3* and *Figure 3—figure supplement 1*.

- Supplementary file 7. Stellaris-designed fluorescence in situ hybridization (FISH) probes specific to unspliced (US) human immunodeficiency virus (HIV) RNA. Related to *Figures 4* and *5*, *Figure 4—figure supplements 1–3*.

- Supplementary file 8. Host protein immunofluorescence in cells with and without siRNA knockdown (KD). Related to *Figure 4* and *Figure 4—figure supplement 1*.

## Data availability

The mass spectrometric data generated during this study are available at MassIVE (https://doi.org/10.25345/C52T8F). Immunofluorescence and western blot raw files are available at Mendeley Data (https://doi.org/10.17632/gf3k5chdff.2). All other data are included with this published article.

The following datasets were generated:

| Author(s) | Year | Dataset title | Dataset URL | Database and Identifier |
|---|---|---|---|---|
| Knoener RA, Edward LE, Becker JT, Scalf M, Benner BE, Sherer NM, Smith LM | 2021 | Identifying HIV RNA splice variant protein interactomes using HyPRMS | https://doi.org/10.25345/C52T8F | MassIVE, 10.25345/C52T8F |
| Knoener R | 2021 | Identification of host proteins differentially associated with HIV-1 RNA splice variants | https://doi.org/10.17632/gf3k5chdff.2 | Mendeley Data, 10.17632/gf3k5chdff.2 |

The following previously published datasets were used:

| Author(s) | Year | Dataset title | Dataset URL | Database and Identifier |
|---|---|---|---|---|
| Queiroz RML, Smith T, Villanueva E, Marti-Solano M, Monti M, Pizzinga M, Mirea DM, Ramakrishna M, Harvey RF, Dezi V | 2019 | Orthogonal Organic Phase Separation protocol | http://proteomecentral.proteomexchange.org/cgi/GetDataset?ID=PXD009668 | ProteomeXchange, PXD009668 |

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

# Appendix 1

**Appendix 1—key resources table**

| Reagent type (species) or resource | Designation | Source or reference | Identifiers | Additional information |
|---|---|---|---|---|
| Cell line (*Homo sapiens*) | Jurkat, Clone E6-1 | NIH AIDS Reagent Program, Division of AIDS, NIAID, NIH | Cat#177; RRID: CVCL_0367 | Male |
| Cell line (*Homo sapiens*) | HEK293T | ATCC | CRL-11268; RRID:CVCL_1926 | Fetus |
| Cell line (*Homo sapiens*) | HEK293T YFP-ACT | ATCC | CRL-11268; RRID:CVCL_1926 | HEK293T cells stably expressing YFP-ACT fusion protein |
| Cell line (*Homo sapiens*) | HeLa | ATCC | CCL-2; RRID: CVCL_0030 | Female |
| Recombinant DNA reagent | HIV-1 E-R- Gag-3x CFP mCherry/nef (plasmid) | *Knoener et al., 2017* | | NL4-3 |
| Recombinant DNA reagent | HIV-1 E-R-CFP (plasmid) | *Knoener et al., 2017* | | NL4-3 |
| Recombinant DNA reagent | psPAX2 (plasmid) | | RRID: Addgene_12660 | A gift from Didier Trono |
| Sequence-based reagent | Intron-1 capture oligonucleotide | This paper, *Supplementary file 1* | | Biotinylated; *Supplementary file 1* |
| Sequence-based reagent | Intron-2 capture oligonucleotide | This paper, | | Biotinylated; *Supplementary file 1* |
| Sequence-based reagent | 3'-Exon capture oligonucleotide | This paper, *Supplementary file 1* | | Biotinylated; *Supplementary file 1* |
| Sequence-based reagent | Intron-1 release oligonucleotide | This paper, *Supplementary file 1* | | *Supplementary file 1* |
| Sequence-based reagent | Intron-2 release oligonucleotide | This paper, *Supplementary file 1* | | *Supplementary file 1* |
| Sequence-based reagent | 3'-Exon release oligonucleotide | This paper, *Supplementary file 1* | | *Supplementary file 1* |
| Sequence-based reagent | Intron-1 qPCR assay | This paper, *Supplementary file 1* | | *Supplementary file 1* |
| Sequence-based reagent | Intron-2 qPCR assay | This paper, *Supplementary file 1* | | *Supplementary file 1* |
| Sequence-based reagent | Intron-3 qPCR assay | This paper, *Supplementary file 1* | | *Supplementary file 1* |
| Sequence-based reagent | RNA FISH probes | Biosearch Technologies; This paper, *Supplementary file 7* | | *Supplementary file 7* |
| Transfected construct (*Homo sapiens*) | siRNAs for knockdown (KD) screen | This paper, *Supplementary file 3* | | *Supplementary file 3* |
| Antibody | Anti-BUB3 (mouse polyclonal) | ThermoFisher | PA5-20388; RRID:AB_11154661 | IF (1:200) WB (1:1000) |
| Antibody | Anti-CSDE1 (rabbit polyclonal) | ThermoFisher | PA5-22394; RRID:AB_11154127 | IF (1:200) WB (1:1000) |

*Continued on next page*

*Appendix 1—key resources table continued*

| Reagent type (species) or resource | Designation | Source or reference | Identifiers | Additional information |
|---|---|---|---|---|
| Antibody | Anti-DHX30 (rabbit polyclonal) | ThermoFisher | PA5-41298; RRID:AB_2607114 | IF (1:200) WB (1:1000) |
| Antibody | Anti-DLD (rabbit polyclonal) | ThermoFisher | PA5-27367; RRID:AB_2544843 | IF (1:200) WB (1:1000) |
| Antibody | Anti-DNM2 (rabbit polyclonal) | ThermoFisher | PA1-661; RRID:AB_2293040 | IF (1:200) WB (1:1000) |
| Antibody | Anti-DYNC1H1 (rabbit polyclonal) | ThermoFisher | PA5-49451; RRID:AB_2634905 | IF (1:200) WB (1:1000) |
| Antibody | Anti-FAM120A (rabbit polyclonal) | ThermoFisher | PA5-54069; RRID:AB_2641236 | IF (1:200) WB (1:1000) |
| Antibody | Anti-G3BP1 (mouse monoclonal) | Santa Cruz | SC-98561; RRID:AB_2294329 | IF (1:200) WB (1:1000) |
| Antibody | Anti-GSDMA (rabbit polyclonal) | ThermoFisher | PA5-24813; RRID:AB_2542313 | IF (1:200) WB (1:1000) |
| Antibody | Anti-HNRNPR (rabbit polyclonal) | ThermoFisher | PA5-55290; RRID:AB_2642500 | IF (1:200) WB (1:1000) |
| Antibody | Anti-IGF2BP3 (rabbit polyclonal) | ThermoFisher | PA5-51672; RRID:AB_2642656 | IF (1:200) WB (1:1000) |
| Antibody | Anti-LRPPRC (rabbit polyclonal) | ThermoFisher | PA5-22034; RRID:AB_11153345 | IF (1:200) WB (1:1000) |
| Antibody | Anti-MBOAT7 (rabbit polyclonal) | Abcam | ab105643; RRID:AB_10862084 | a.k.a. Anti-LENG4 IF (1:200) WB (1:1000) |
| Antibody | Anti-NCLN (rabbit polyclonal) | ThermoFisher | PA5-34356; RRID:AB_2551708 | IF (1:200) WB (1:1000) |
| Antibody | Anti-RBM4 (rabbit polyclonal) | ThermoFisher | PA5-21755; RRID:AB_11153613 | IF (1:200) WB (1:1000) |
| Antibody | Anti-RBMX (rabbit polyclonal) | ThermoFisher | PA5-49468; AB_2634922 | IF (1:200) WB (1:1000) |
| Antibody | Anti-RPL15 (rabbit polyclonal) | ThermoFisher | PA5-48446; RRID:AB_2633903 | IF (1:200) WB (1:1000) |
| Antibody | Anti-SRRM2 (rabbit polyclonal) | ThermoFisher | PA5-59559; RRID:AB_2647934 | IF (1:200) WB (1:1000) |
| Antibody | Anti-SRSF6 (rabbit polyclonal) | ThermoFisher | PA5-56034; RRID:AB_2647943 | IF (1:200) WB (1:1000) |
| Antibody | Anti-TRIM56 (mouse monoclonal) | ThermoFisher | MA5-27066; RRID:AB_2725573 | IF (1:200) WB (1:1000) |

*Appendix 1—key resources table continued*

| Reagent type (species) or resource | Designation | Source or reference | Identifiers | Additional information |
|---|---|---|---|---|
| Antibody | Anti-HIV-1 Gag/p24 | NIH AIDS Reagent Program | 183-H12-5C; RRID:AB_2819250 | IF (1:200) WB (1:1000) |
| Antibody | Goat anti-Rabbit secondary antibody | ThermoFisher | A11008; RRID:AB_143165 | Conjugate Alexa Fluor 488 IF (1:200) WB (1:10000) |
| Antibody | Goat anti-Mouse secondary antibody | ThermoFisher | A21235; RRID:AB_2535804 | Conjugate Alexa Fluor 647 IF (1:200) WB (1:10000) |
| Antibody | Goat anti-Rabbit secondary antibody | LiCor Biosciences | 926–32211; RRID:AB_621843 | LiCor IRDye800 WB (1:10000) |
| Antibody | Goat anti-Mouse secondary antibody | LiCor Biosciences | 926–68020; RRID:AB_10706161 | LiCor 680LT WB (1:10000) |
| Commercial assay or kit | High Capacity cDNA Reverse Transcription Kit | ThermoFisher | 4368814 | |
| Peptide, recombinant protein | Trypsin | Promega | V5111 | |
| Peptide, recombinant protein | Proteinase K | Sigma | 3115828001 | |
| Peptide, recombinant protein | RNasin Plus | Promega | N2611 | |
| Chemical compound, drug | Halt Protease Inhibitors | ThermoFisher | 78430 | |
| Chemical compound, drug | Tri Reagent | Sigma | T9424 | |
| Other | GlycoBlue | ThermoFisher | AM9516 | |
| Other | OMIX C18 solid-phase extraction pipette tip | Agilent | A57009100 | |
| Other | DarmaFECT | Horizon Discovery | T-2001–02 | |
| Other | DAPI | Sigma | D9542-10MG | |
| Other | FISH wash buffer A | Biosearch Technologies | SMF-WA1-60 | |
| Other | FISH wash buffer B | Biosearch Technologies | SMF-WB1-20 | |
| Other | FISH hybridization buffer | Biosearch Technologies | SMF-HB1-10 | |
| Other | TaqMan Fast Advanced Master Mix | Fisher Scientific | 4444963 | |
| Other | Ribonucleoside Vanadyl Complex | Sigma | R3380-5ML | |
| Other | Sera-Mag Streptavidin Coated Magnetic Speedbeads | Fisher Scientific | 09981140 | |
| Software, algorithm | MaxQuant software | https://www.maxquant.org/ | RRID:SCR_014485 | |

*Continued on next page*

*Appendix 1—key resources table continued*

| Reagent type (species) or resource | Designation | Source or reference | Identifiers | Additional information |
|---|---|---|---|---|
| Software, algorithm | Perseus software | https://www.maxquant.org/perseus/ | RRID:SCR_015753 | |
| Software, algorithm | Cluster software | http://bonsai.hgc.jp/~mdehoon/software/cluster/software.htm | | |
| Software, algorithm | TreeView software | https://sourceforge.net/projects/jtreeview/ | RRID:SCR_016916 | |
| Software, algorithm | Gene Ontology software | http://geneontology.org/ | RRID:SCR_002811 | |
| Software, algorithm | FIJI/ImageJ2 software | https://imagej.nih.gov/ij/ | RRID:SCR_003070 | |

