## [Decision Letter]

**Acceptance summary:**

The manuscript describes a new method to identify proteins that interact with different splice variants of HIV-1 RNA, called HyPR-MS_SV_ (Hybridization Purification of RNA-Protein Complexes Followed by Mass Spectrometry for splice variants), and its application to identify cellular RNA binding proteins that interact with each of three major HIV-1 RNA species: unspliced RNA, partially spliced RNA, and completely spliced RNA. The authors also present a comprehensive study of subcellular localization, co-localization with other viral factors, and results of knockdowns of identified host factors to further reveal their potential roles in HIV-1 RNA biology. This technical tour-de-force will serve as a valuable resource, and should prompt further mechanistic studies of host factors that influence RNA trafficking and HIV-1 infection.

**Decision letter after peer review:**

Thank you for submitting your article "Identifying HIV-1 RNA splice variant protein interactomes using HyPR-MS_SV_" for consideration by *eLife*. Your article has been reviewed by three peer reviewers, and the evaluation has been overseen by a Reviewing Editor and Detlef Weigel as the Senior Editor. The reviewers have opted to remain anonymous.

The reviewers have discussed the reviews with one another and the Reviewing Editor has drafted this decision to help you prepare a revised submission.

Knoener et al. describe a new method for identifying proteins that interact with different splice variants of HIV-1 RNA, called HyPR-MS_SV_ (Hybridization Purification of RNA-Protein Complexes Followed by Mass Spectrometry for splice variants). The method was developed to identify cellular RNA binding proteins that bind to each of three major HIV-1 RNA species: unspliced RNA, partially spliced RNA, and completely spliced RNA. The authors present a comprehensive study of subcellular localization, co-localization with other viral factors, and results of knockdowns of identified host factors to reveal potential roles in HIV-1 RNA biology. This is a technical tour-de-force, the experiments are rigorously performed, and the results are presented in detail. A limitation is the lack of virology (e.g., testing for loss of viral infectivity upon factor knockdown), but as a methodology paper this is valuable resource, and the results will lead to further studies that will add to improved understanding about host factors that influence RNA trafficking and function in HIV-1 infection.

Essential revisions:

1) The probe used for CS RNA is within the 3' exon that is present in all 3 RNA species (US, CS, and PS) yet the analysis was performed in a pairwise fashion (CS vs. PS and US vs. CS). Were there host factors that bound to all 3 RNA species? if so, please indicate them and provide an explanation for why this may be so. Are there any RNA binding proteins that have been shown previously to interact non-specifically to RNA that could potentially be discounted?

2) Did the authors note that several factors were omitted from their identified protein list that have been previously shown to have an effect on HIV-1 RNA or were packaged into virus particles? Factor that comes to mind include DHX9 (RHA), DDX3, eIF4E, PABP, and nuclear cap binding complex, which are missing from the analysis and have been shown to be important in HIV biology. It would be informative if the authors would comment on why factors previously shown to be important were not found in their study?

3) A previous study that examined host factors that bind to unspliced HIV-1 RNA (5'UTR) was not cited, although some of the factors in that paper and this work are overlapping (HNRNPR, SRSF6, IGF2BP3, SYNCRIP, YBX1 as examples) (Stake et al., 2015). In a similar vein, the authors should briefly discuss how their HIV RNA-protein interactome compare to other studies that used segments of the genome as bait or where affinity handles were engineered in.

4) The HYPER-MS_SV_ studies were performed in Jurkat cells while the validation experiment were carried out in 293T or He La cells. While this is acceptable given the questions being asked, the different cell lines might differentially express other factors that influence the activity of the protein being KO or IF. The authors should clarify whether they consider this a limitation of the study and what interpretations it might impact.

5) Figure 2B shows that the largest group of cellular proteins that preferentially interact with US viral transcripts belong to carboxylic acid and coenzyme metabolic pathways. This is an interesting observation that the authors did not validate or discuss a in terms of a potential virological function. Although we will not require it, the authors could consider performing similar KO/FISH experiments to better understand how a subset of those metabolic enzymes regulate HIV gene expression.

---

## [Author Response]

Essential revisions:1) The probe used for CS RNA is within the 3' exon that is present in all 3 RNA species (US, CS, and PS) yet the analysis was performed in a pairwise fashion (CS vs. PS and US vs. CS). Were there host factors that bound to all 3 RNA species? if so, please indicate them and provide an explanation for why this may be so. Are there any RNA binding proteins that have been shown previously to interact non-specifically to RNA that could potentially be discounted?

We thank the reviewers for this important comment. Indeed, we identified over 900 host factors associated with all three splice variant pools. We agree that including these factors and discussing their significance and specificity to HIV-1 RNA would be valuable. To this end, we made the following changes to the revised manuscript:

1) We constructed a list of 926 proteins identified in at least two biological replicates for all three splice variant capture types (US, PS, and CS). This list and the data are included as a tab named “Common Protein Interactors” in Supplementary file 2.

2) We mined the literature for prior studies that used PolyA-tail capture strategies to identify “common” cellular RNA-associated proteins and then annotated our “Common Protein Interactors” tab in Supplementary file 2 to highlight which of the proteins identified in our HIV-1 study are most likely to represent “non-specific” mRNA-associated proteins.

3) We also added a tab named “Differential Protein Interactors” to Supplementary file 2 that highlights which of the 212 proteins identified as differential HIV-1 splice variant interactors in our study were previously identified as more general RNA-associated proteins in the studies noted above.

4) Finally, we modified the text to discuss these data in the Results and Discussion sections (see below).

“In all, 926 proteins were identified in at least two biological replicates of all three HIV splice variant captures (Supplementary file 2). […] Furthermore, several common interactors matched proteins previously identified in RNA capture screens that had used partial segments of HIV-1 RNA as bait, including 27 of 32 proteins identified by Kula, et al., 32 of the 41 identified by Marchand, et al., and 93 of the 121 identified by Stake, et al. (Supplementary file 2) (Kula et al., 2011; Marchand et al., 2011; Stake et al., 2015).”

“Over 900 proteins were detected in the interactomes of all three HIV splice variant classes (Supplementary file 2); many of these are likely “general” RNA regulatory factors as they were found associated with cellular polyadenylated RNAs in prior studies (Hentze et al., 2018; Queiroz et al., 2019). However, many of these common interactors included RNA regulatory proteins already demonstrated to play important roles in HIV-1 replication (e.g., DHX9, DDX3, SR proteins, and ABCE1) (Boeras et al., 2016; Bolinger et al., 2010; Friedrich et al., 2011; Mahiet and Swanson, 2016; Soto-Rifo et al., 2013; Stoltzfus and Madsen, 2006; Yedavalli et al., 2004) or were identified in prior pull-down studies (Kula et al., 2011; Marchand et al., 2011; Stake et al., 2015).”

2) Did the authors note that several factors were omitted from their identified protein list that have been previously shown to have an effect on HIV-1 RNA or were packaged into virus particles? Factor that comes to mind include DHX9 (RHA), DDX3, eIF4E, PABP, and nuclear cap binding complex, which are missing from the analysis and have been shown to be important in HIV biology. It would be informative if the authors would comment on why factors previously shown to be important were not found in their study?

We agree that we did not include sufficient description or discussion of known HIV-1 RNA host co-factors and regret this deficiency. Many of the known factors noted by the reviewers (e.g., DHX9, DDX3, PABP, NCBP1 (component of the nuclear cap-binding complex)) were identified as common protein interactors; and they, among other known host regulatory factors, are now each featured in Supplementary file 2 (see also response to Essential revisions #1). We also added a more thorough discussion of these proteins and other known HIV-1 RNA effectors to the Results and Discussion sections (see below).

“These common interactors also featured several proteins previously implicated in HIV-1 replication including host factors regulating RNA transport and translational initation (e.g., NCBP1, DHX9, DDX3, EIF4G, and PABP) (Boeras et al., 2016; Bolinger et al., 2010; Soto-Rifo et al., 2013; Stake et al., 2015; Yedavalli et al., 2004), known HIV splicing factors (e.g., HMGA1, HNRNPA1, HNRNPAB, HNRNPH, HNRNPF, SRSF1, SRSF2, SRSF3, SRSF6, SRSF7, TRA2B, and U2AF2) (Dlamini and Hull, 2017; Mahiet and Swanson, 2016; Sertznig et al., 2018; Stoltzfus and Madsen, 2006), RNA nuclear export and transport proteins (e.g., ABCE1, RAB11A, RANBP2, and XPO1) (Friedrich et al., 2011) and proteins implicated in HIV-1 virus particle assembly (e.g., AP-2, PDCD6IP (ALIX), STAU2, UPF1, and VPS4) (Friedrich et al., 2011; Meng and Lever, 2013). Furthermore, several common interactors matched proteins previously identified in RNA capture screens that had used partial segments of HIV-1 RNA as bait, including 27 of 32 proteins identified by Kula, et al., 32 of the 41 identified by Marchand, et al., and 93 of the 121 identified by Stake, et al. (Supplementary file 2)”.

“However, many of these common interactors included RNA regulatory proteins already demonstrated to play important roles in HIV-1 replication (e.g., DHX9, DDX3, SR proteins, and ABCE1) (Boeras et al., 2016; Bolinger et al., 2010; Friedrich et al., 2011; Mahiet and Swanson, 2016; Soto-Rifo et al., 2013; Stoltzfus and Madsen, 2006; Yedavalli et al., 2004) or were identified in prior pull-down studies (Kula et al., 2011; Marchand et al., 2011; Stake et al., 2015).”

3) A previous study that examined host factors that bind to unspliced HIV-1 RNA (5'UTR) was not cited, although some of the factors in that paper and this work are overlapping (HNRNPR, SRSF6, IGF2BP3, SYNCRIP, YBX1 as examples) (Stake et al., 2015). In a similar vein, the authors should briefly discuss how their HIV RNA-protein interactome compare to other studies that used segments of the genome as bait or where affinity handles were engineered in.

We apologize for not citing the Stake et al. paper (Stake et al., 2015) considering its importance to our study. This was a mistake that occurred during editing and we have remedied the oversight. We also appreciate the reviewers’ request that we better compare our findings to those of others that used HIV RNA baits in different contexts. To do so, we now cross-reference both our differential and common protein interactor lists in Supplementary file 2 to the lists of proteins identified by Stake et al. and two additional studies (Kula et al; and Marchand et al.) and indicate which proteins were common among them. We also added discussion of these studies to the Results and Discussion sections (see below).

“Furthermore, several common interactors matched proteins previously identified in RNA capture screens that had used partial segments of HIV-1 RNA as bait, including 27 of 32 proteins identified by Kula, et al., 32 of the 41 identified by Marchand, et al., and 93 of the 121 identified by Stake, et al. (Supplementary file 2) (Kula et al., 2011; Marchand et al., 2011; Stake et al., 2015).”

“To our knowledge, all previous studies for discovery of HIV RNA protein interactors, with the exception of our prior study (Knoener et al., 2017), utilized synthetic viral RNAs as bait added to cellular lysates (Marchand et al., 2011; Singh et al., 2016; Stake et al., 2015) or viral constructs engineered to encode artificial RNA sequences for the purpose of RNA “tagging” (e.g., MS2 loops) (Kula et al., 2011). […] Furthermore, this strategy can, in theory, be expanded to differentiate the protein interactomes of each individual protein-coding HIV-1 RNA, can be used to extract native RNA transcripts produced from any strain or infected cell type, and can easily be adapted to study other viruses or cellular RNA splice variants.”

4) The HYPER-MS_SV_ studies were performed in Jurkat cells while the validation experiment were carried out in 293T or He La cells. While this is acceptable given the questions being asked, the different cell lines might differentially express other factors that influence the activity of the protein being KO or IF. The authors should clarify whether they consider this a limitation of the study and what interpretations it might impact.

We appreciate the reviewers’ comment and now detail in both the Results and Discussion sections (see below) the reasons for selecting these cell lines for each application and acknowledge the limitations of these choices. Although using highly tractable cell systems was invaluable for establishing and validating the HyPR-MS_SV_ technology, it is certainly our intention that the strategy next be applied to study more relevant R5-tropic HIV-1 founder strains primary CD4^+^ T cells and monocytes/macrophages.

“The Jurkat cell line was chosen because it is a well characterized CD4^+^ T cell line previously confirmed to support replication of the HIV-1_NL4-3_ reporter virus used for this study (Knoener et al., 2017).”

“The HEK293T cell line was selected here due to its extensive use in studies of HIV-1 expression and its compatibility with siRNA transfection experiments (Konig et al., 2008).”

“HeLa cells were chosen based on their microscopy-conducive size and shape (large and flat) and their common use to study HIV-1 gene expression (Jouvenet et al., 2008; Pocock et al., 2016).”

“It should be noted, the HyPR-MS interactome determination, siRNA knockdown screen and RNA-protein co-localization and expression analysis were completed using three different cell lines; Jurkat, 293T, and HeLa, respectively. […] To overcome some of these limitations, future work will include conducting similar screens in primary CD4^+^ T cells and monocytes/macrophages using more relevant R5-tropic HIV-1 founder strains for elucidation of innate immune signaling host factors involved in HIV-1 RNA biology.”

5) Figure 2B shows that the largest group of cellular proteins that preferentially interact with US viral transcripts belong to carboxylic acid and coenzyme metabolic pathways. This is an interesting observation that the authors did not validate or discuss a in terms of a potential virological function. Although we will not require it, the authors could consider performing similar KO/FISH experiments to better understand how a subset of those metabolic enzymes regulate HIV gene expression.

This is an interesting point and we have added text to both the Results and Discussion sections (see below) that better highlight these findings and briefly discuss what has recently been discovered with regards to metabolic proteins having functions as RNA-binding proteins. We have also now added these proteins in Figure 4C so that readers can readily interrogate the lists. We agree that pursuing this category of proteins for further validation and functional analysis will be an important future avenue of study.

“…and several have known roles in the carboxylic acid metabolic process, a GO term also over-represented in the US RNA interactome (Figure 2C, Figure 2—source data 2).”

“Notably, many metabolic and enzymatic proteins were identified as US HIV RNA interactors (Figure 2B and C). […] This suggests that metabolic proteins may play an important role in HIV gene expression, but further investigation is need to decipher the unique role of each.”